# Face Reconstruction from Facial Templates by Learning Latent Space of a Generator Network

**Hatef Otroshi Shahreza**[1,2] **and Sébastien Marcel**[1,3]
[1]Idiap Research Institute, Martigny, Switzerland
[2]École Polytechnique Fédérale de Lausanne (EPFL), Lausanne, Switzerland
[3]Université de Lausanne (UNIL), Lausanne, Switzerland
`{hatef.otroshi,sebastien.marcel}@idiap.ch`

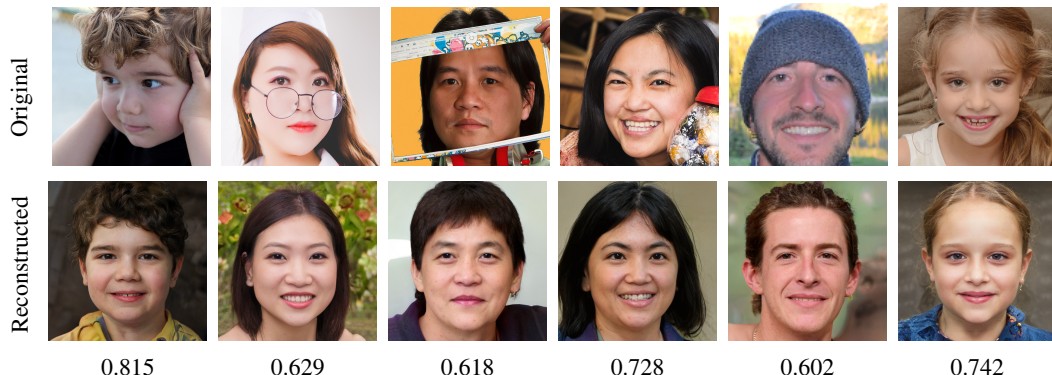

| | | | | | |
|---|---|---|---|---|---|
| 0.815 | 0.629 | 0.618 | 0.728 | 0.602 | 0.742 |

Figure 1: Sample face images from the FFHQ dataset and their corresponding reconstructed images using our template inversion method from ArcFace templates (used in a face recognition system). The values below each image show the cosine similarity between the corresponding templates of original and reconstructed face images.

## Abstract

In this paper, we focus on the template inversion attack against face recognition systems and propose a new method to reconstruct face images from facial templates. Within a generative adversarial network (GAN)-based framework, we learn a mapping from facial templates to the intermediate latent space of a pre-trained face generation network, from which we can generate high-resolution realistic reconstructed face images. We show that our proposed method can be applied in whitebox and blackbox attacks against face recognition systems. Furthermore, we evaluate the transferability of our attack when the adversary uses the reconstructed face image to impersonate the underlying subject in an attack against another face recognition system. Considering the adversary's knowledge and the target face recognition system, we define five different attacks and evaluate the vulnerability of state-of-the-art face recognition systems. Our experiments show that our proposed method achieves high success attack rates in whitebox and blackbox scenarios. Furthermore, the reconstructed face images are transferable and can be used to enter target face recognition systems with a different feature extractor model. We also explore important areas in the reconstructed face images that can fool the target face recognition system.

37th Conference on Neural Information Processing Systems (NeurIPS 2023).

# 1 Introduction

Face recognition (FR) systems tend toward ubiquity, and their applications, which range from cell phone unlock to national identity system, border control, etc., are growing rapidly. Typically, in such systems, a feature vector (called embedding or template) is extracted from each face image using a deep neural network, and is stored in the system's database during the enrollment stage. During the recognition stage, either verification or identification, the extracted feature vector is compared with the ones in the system's database to measure the similarity of identities. Among potential attacks against FR systems [Galbally et al., 2014, Biggio et al., 2015, Hadid et al., 2015, Mai et al., 2018, Marcel et al., 2023], the template inversion (TI) attack significantly jeopardizes the users' privacy. In a TI attack, the adversary gains access to templates stored in the FR system's database and aims to reconstruct the underlying face image. Then, the adversary not only achieves privacy-sensitive information (such as gender, ethnicity, etc.) of enrolled users, but also can use reconstructed face images to impersonate.

In this paper, we focus on the TI attack against FR systems and propose a novel method to reconstruct face images from facial templates (Figure 1 shows sample reconstructed face images using our proposed method). Within a generative adversarial network (GAN)-based framework, we learn a mapping from face templates to the intermediate latent space of StyleGAN3 [Karras et al., 2021], as a pre-trained face generation network. Then, using the synthesis part of StyleGAN3, we can generate high-resolution realistic face image. Our proposed method can be applied for *whitebox* and *blackbox* attacks against FR systems. In the *whitebox* scenario, the adversary knows the internal functioning of the feature extraction model and its parameters. However, in the *blackbox* scenario, the adversary does not know the internal functioning of the feature extraction model and can only use it to extract features from any arbitrary image. Instead, we assume that the adversary has a *whitebox* of another FR model, which can be used for training the face reconstruction network. We also evaluate the transferability of our attack by considering the case where the adversary uses the reconstructed face image to impersonate the underlying subject in an attack against another FR system (which has a different feature extraction model). Considering the adversary's knowledge and the target FR system, we define five different attacks, and evaluate the vulnerability of state-of-the-art (SOTA) FR systems. Figure 2 illustrates the general black diagram of our proposed template inversion attack.

To elaborate on the contributions of our paper, we list them hereunder:

- We propose a novel method to generate high-resolution realistic face images from facial templates. Within a GAN-based framework, we learn the mapping from facial templates to the latent space of a pre-trained face generation network.

- We propose our method for *whitebox* and *blackbox* scenarios. While our method is based on the *whitebox* knowledge of the FR model, we extend our attack *blackbox* scenario, using another FR model that the adversary has access to.

- We define five different attacks against FR systems (based on the adversary's knowledge and the target system), and evaluate the transferability of the reconstructed face images and vulnerability of SOTA FR models to TI attacks. We also explore important areas in the reconstructed face images. To our knowledge, this is the first work which comprehensively evaluates the transferability of the reconstructed face images in TI attacks.

The remainder of the paper is organized as follows: Section 2 introduces the problem formulation and our proposed face reconstruction method. Section 3 covers the related works in the literature and

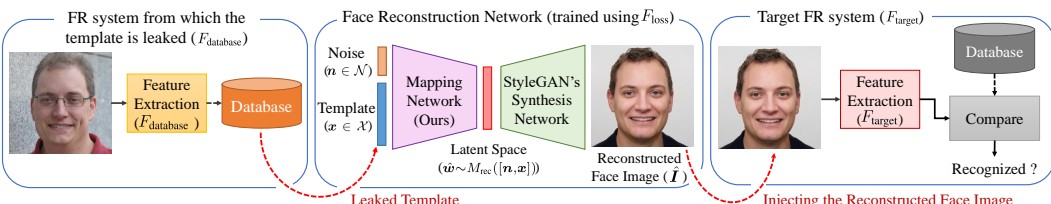

Figure 2: Block diagram of our proposed template inversion attack

compares them with our proposed method. Section 4 presents our experiential results. Finally, the paper is concluded in Section 5.

## 2 Problem Definition and Proposed Method

In this paper, we consider a TI attack against a FR system based on the following threat model:

- *Adversary's goal*: The adversary aims to reconstruct a face image from a template, and use the reconstructed face image to enter the same or a different face recognition system, which we call the target FR system.

- *Adversary's knowledge:* The adversary knows a face template of a user enrolled in the FR system's database. The adversary also has either *whitebox* or *blackbox* knowledge of the feature extractor model in the same FR system.

- *Adversary's capability:* The adversary can present the reconstructed face image to the target FR system (e.g., using a printed photograph). However, for simplicity, we consider that adversary can inject the reconstructed face image as a query to the target FR system.

- *Adversary's strategy:* The adversary can train a face reconstruction model to invert facial templates and reconstruct underlying face images. Then, the adversary can use the reconstructed face images to inject as a query to the target FR system, to enter that system.

Let $F(.)$ denote a facial feature extraction model, which gets the face image $\boldsymbol{I} \in \mathcal{I}$ and extracts facial template $\boldsymbol{x} = F(\boldsymbol{I}) \in \mathcal{X}$. According to the threat model, the adversary has access to the target facial template $\boldsymbol{x}_{\text{database}} = F_{\text{database}}(\boldsymbol{I})$ and aims to generate a reconstructed face image $\hat{\boldsymbol{I}}$. Then, the adversary can use the reconstructed face image $\hat{\boldsymbol{I}}$ to impersonate the corresponding subject and attack a target FR system with $F_{\text{target}}(.)$, which might be different from $F_{\text{database}}(.)$.

To train a face reconstruction model, we can use a dataset of face images $\{\boldsymbol{I}_i\}_{i=1}^N$ with $N$ face images (no label is required), and generate a training dataset $\{(\boldsymbol{x}_i, \boldsymbol{I}_i)\}_{i=1}^N$, where $\boldsymbol{x}_i = F_{\text{database}}(\boldsymbol{I}_i)$. Then, a face reconstruction model $G(.)$ can be trained to reconstruct face image $\hat{\boldsymbol{I}} = G(\boldsymbol{x})$ given each facial template $\boldsymbol{x} \in \mathcal{X}$. To train such a face reconstruction model, we consider a multi-term face reconstruction loss function as follows:

$$\mathcal{L}_{\text{rec}} = \mathcal{L}_{\text{pixel}} + \mathcal{L}_{\text{ID}}, \tag{1}$$

where $\mathcal{L}_{\text{pixel}}$ and $\mathcal{L}_{\text{ID}}$ indicate pixel loss and ID losses, respectively, and are defined as:

$$\mathcal{L}_{\text{pixel}} = \mathbb{E}_{\boldsymbol{x} \sim \mathcal{X}}[\|\boldsymbol{I} - G(\boldsymbol{x})\|_2^2], \tag{2}$$

$$\mathcal{L}_{\text{ID}} = \mathbb{E}_{\boldsymbol{x} \sim \mathcal{X}}[\|F_{\text{loss}}(\boldsymbol{I}) - F_{\text{loss}}(G(\boldsymbol{x}))\|_2^2]. \tag{3}$$

The pixel loss is used to minimize the pixel-level reconstruction error of the generated face image. The ID loss is also used to minimize the distance between facial templates extracted by $F_{\text{loss}}(.)$ from original and reconstructed face images. In Eq. 3, $F_{\text{loss}}(.)$ denotes a feature extraction model that the adversary is assumed to have complete knowledge of its parameters and internal functioning. Based on the adversary's knowledge of $F_{\text{database}}(.)$ (i.e., *whitebox* or *blackbox* scenarios), $F_{\text{loss}}(.)$ might be the same or different from $F_{\text{database}}(.)$.

For the face reconstruction model, we consider StyleGAN3 [Karras et al., 2021], as a pre-trained face generation network[1]. The StyleGAN3 model is trained on a dataset of face images using a GAN-based framework that can generate high-resolution and realistic face images. The structure of StyleGAN3 is composed of two networks, mapping and synthesis networks. The mapping network $M_{\text{StyleGAN}}(.)$ gets a random noise $\boldsymbol{z} \in \mathcal{Z}$ and generates an intermediate latent code $\boldsymbol{w} = M_{\text{StyleGAN}}(\boldsymbol{z}) \in \mathcal{W}$. Then, the latent code $\boldsymbol{w}$ is given to the synthesis network $S_{\text{StyleGAN}}(.)$ to generate a face image. In our training process, we fix the synthetic network $S_{\text{StyleGAN}}(.)$ and train a new mapping $M_{\text{rec}}(.)$ to generate $\hat{\boldsymbol{w}}$ corresponding to the given facial template $\boldsymbol{x} \in \mathcal{X}$. Then, the generated latent code $\hat{\boldsymbol{w}}$ is given to the synthesis network $S_{\text{StyleGAN}}(.)$ to generate the reconstructed face image $\hat{\boldsymbol{I}} = S_{\text{StyleGAN}}(\hat{\boldsymbol{w}})$. We can

---

[1]While we use StyleGAN3 in our experiments, our method can also be used with other face generator networks. In our experiment in Section A.3 of the appendix, we use StyleSwin [Zhang et al., 2022] as the face generator network and Figure 7 of the appendix shows reconstructed face images of our method using StyleSwin.

train our new mapping $M_{\text{rec}}(.)$ using our reconstruction loss function as in Eq. 1. However, to obtain a realistic face image from the generated $\hat{w}$ through the pre-trained synthetic network $S_{\text{StyleGAN}}(.)$, the generated $\hat{w}$ needs to be in the distribution $\mathcal{W}$; otherwise, the output may not look like a real human face. Hence, to generate $\hat{w}$ vectors such that they have the same distribution as StyleGAN's intermediate latent, $w \in \mathcal{W}$, we use a GAN-based framework to learn the distribution $\mathcal{W}$. To this end, we use the Wasserstein GAN (WGAN) [Arjovsky et al., 2017] algorithm to train a critic network $C(.)$ which critiques the generated $\hat{w}$ vectors compared to the real StyleGAN's $w \in \mathcal{W}$ vectors, and simultaneously we optimize our mapping network to generate $\hat{w}$ vectors with the same distribution as $\mathcal{W}$. Hence, we can consider our mapping network $M_{\text{rec}}(.)$ as a conditional generator in our WGAN framework, which generates $\hat{w} = M_{\text{rec}}([\boldsymbol{n}, \boldsymbol{x}])$ given a facial template $x \in \mathcal{X}$ and a random noise vector $\boldsymbol{n} \in \mathcal{N}$. Then, we can train our mapping network and critic network using the following loss functions:

$$\mathcal{L}_C^{\text{WGAN}} = \mathbb{E}_{\boldsymbol{w} \sim \mathcal{W}}[C(\boldsymbol{w})] - \mathbb{E}_{\hat{\boldsymbol{w}} \sim M_{\text{rec}}([\boldsymbol{n}, \boldsymbol{x}])}[C(\hat{\boldsymbol{w}})] \qquad (4)$$

$$\mathcal{L}_{M_{\text{rec}}}^{\text{WGAN}} = \mathbb{E}_{\hat{\boldsymbol{w}} \sim M_{\text{rec}}([\boldsymbol{n}, \boldsymbol{x}])}[C(\hat{\boldsymbol{w}})] \qquad (5)$$

In a nutshell, we train a new mapping network $M_{\text{rec}}(.)$ using our reconstruction loss function in Eq. 1, and also optimize $M_{\text{rec}}(.)$ within our WGAN framework using Eq. 5. Simultaneously, we also train the critic network $C(.)$ within our WGAN training using Eq. 4 to learn the distribution of StyleGAN's intermediate latent space $\mathcal{W}$ and help our mapping network $M_{\text{rec}}(.)$ to generate vectors with the same distribution as $\mathcal{W}$. Figure 3 depicts the block diagram of the proposed method. We should note that our mapping network $M_{\text{rec}}(.)$ has 2 fully connected layers with Leaky ReLU activation function.

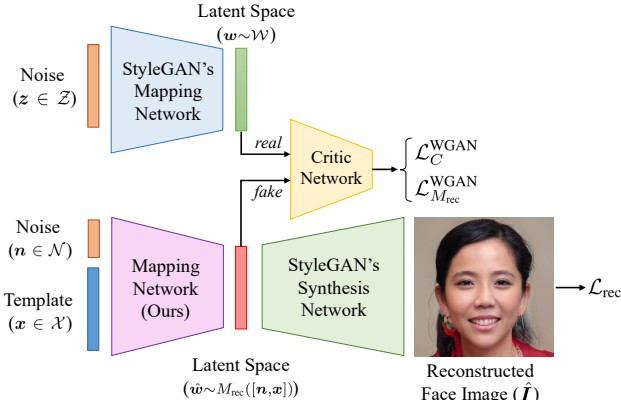

Figure 3: Block diagram of our face reconstruction network.

In our problem formulation, we consider three different feature extraction models, including $F_{\text{database}}(.)$, $F_{\text{loss}}(.)$, and $F_{\text{target}}(.)$. Hence, based on the adversary's knowledge and the target system, we can consider five different attacks:

- **Attack 1:** The adversary has *whitebox* knowledge of the system from which the template is leaked and wants to attack the same system (i.e., $F_{\text{database}} = F_{\text{loss}} = F_{\text{target}}$).

- **Attack 2:** The adversary has *whitebox* knowledge of the feature extractor of the system from which the template is leaked, but aims to attack to a different FR system (i.e., $F_{\text{database}} = F_{\text{loss}} \neq F_{\text{target}}$).

- **Attack 3:** The adversary wants to attack the same system from which the template is leaked, but has only *blackbox* access to the feature extractor of the system. Instead, we assume that the adversary has the *whitebox* knowledge of another FR model to use for training (i.e., $F_{\text{database}} = F_{\text{target}} \neq F_{\text{loss}}$).

- **Attack 4:** The adversary aims to attack a different FR system than the one from which the template is leaked. In addition, the adversary has *whitebox* knowledge of the feature extractor of the target system (i.e., $F_{\text{database}} \neq F_{\text{loss}} = F_{\text{target}}$).

- **Attack 5:** The adversary aims to attack a different FR system from which the template is leaked and has only *blackbox* knowledge of both the target system and the one from which the template is leaked. However, the adversary instead has the *whitebox* knowledge of another FR model to use for training (i.e., $F_{\text{database}} \neq F_{\text{loss}} \neq F_{\text{target}}$).

In the attack 1 and attack 2, the adversary has the *whitebox* knowledge of the system from which the template is leaked (i.e., $F_{\text{database}}(.)$) and uses the same model as $F_{\text{loss}}(.)$ for training the reconstruction network. However, in attacks 3-5, the adversary has the *blackbox* knowledge of the system from which the template is leaked, and therefore uses another FR model as $F_{\text{loss}}(.)$. Comparing the knowledge

Table 1: Comparison of different attacks in this paper.

| | $F_{\text{database}}$ | $F_{\text{loss}}$ | Evaluation | Adversary's Knowledge of Original and Target Systems | Difficulty of Attack |
|---|---|---|---|---|---|
| **Attack 1** | whitebox | $F_{\text{database}}$ | same system | whitebox knowledge of $F_{\text{database}}$ and $F_{\text{target}}$ | very easy |
| **Attack 2** | whitebox | $F_{\text{database}}$ | different system (transferability) | whitebox knowledge of $F_{\text{database}}$ | easy |
| **Attack 3** | blackbox | adversary's own | same system | blackbox knowledge of $F_{\text{database}}$ and $F_{\text{target}}$ | difficult |
| **Attack 4** | blackbox | $F_{\text{target}}$ | different system (transferability) | blackbox knowledge of $F_{\text{database}}$ and whitebox knowledge of $F_{\text{target}}$ | difficult |
| **Attack 5** | blackbox | adversary's own | different system (transferability) | only blackbox knowledge of $F_{\text{database}}$ | very difficult |

of the adversary in these attacks, we expect that attack 1 be the easiest attack for the adversary and attack 5 be the most difficult one. Table 1 compares adversary's knowledge and difficulty of different attacks defined in this paper.

## 3 Related Works

Table 2 compares our proposed method with related works in the literature. Generally, the methods for TI attack against FR systems, can be categorized based on different aspects, including the resolution of generated face images (high/low resolution), the type of attack (*whitebox*/*blackbox* attack), and the basis of the method (optimization/learning-based).

Zhmoginov and Sandler [2016] proposed an optimization-based method and a learning-based method to generate low-resolution face images in the *whitebox* attack against FR systems. In their optimization-based attack, they used a gradient-descent-based approach to find an image that minimizes the distance of the face template as well as some regularization terms to generate a smooth image, including the total variation and Laplacian pyramid gradient normalization [Burt and Adelson, 1987] of the reconstructed face image. In their learning-based attack, they trained a convolutional neural network (CNN) with the same loss terms to generate face images from given facial templates.

Cole et al. [2017] proposed a learning-based attack to generate low-resolution images using a multi-layer perceptron (MLP) to estimate landmark coordinates and a CNN to generate face textures, and then reconstructed face images using a differentiable warping based on estimated landmarks and face texture. They trained their networks in an end-to-end fashion, and minimized the errors for landmark estimation and texture generation as well as the distance of face template as their loss function. To extend their method from the *whitebox* attack to the *blackbox* attack, they proposed not to minimize the distance of face templates in their loss function.

Mai et al. [2018] proposed a learning-based attack to generate low-resolution images in the *blackbox* attack against FR systems. They proposed new convolutional blocks, called neighborly deconvolution blocks A/B (shortly, NbBlock-A and NbBlock-B), and used these blocks to reconstruct face images. They trained their proposed networks using two loss functions, including pixel loss (i.e., $\ell_2$ norm of reconstruction pixel error) and perceptual loss (i.e., $\ell_2$ norm of distance for intermediate features of VGG-19 [Simonyan and Zisserman, 2014] given original and reconstructed face images). They

Table 2: Comparison with related works.

| Reference | Resolution | Basis | White/Black-box | Transferability Eval. | Available code |
|---|---|---|---|---|---|
| Zhmoginov and Sandler [2016] | low | 1) optimization 2) learning | whitebox | ✗ | ✗ |
| Cole et al. [2017] | low | learning | both* | ✗ | ✗ |
| Mai et al. [2018] | low | learning | blackbox | ✗ | ✓ |
| Duong et al. [2020] | low | learning | both** | ✗ | ✗ |
| Truong et al. [2022] | low | learning | both** | ✗ | ✗ |
| Ahmad et al. [2022] | low | learning | blackbox | ✗ | ✗ |
| Dong et al. [2021] | high | learning | blackbox | ✗ | ✓ |
| Vendrow and Vendrow [2021] | high | optimization | blackbox | ✗ | ✓ |
| Dong et al. [2023] | high | optimization | blackbox | ✗ | ✓ |
| Ours | high | learning | both*** | ✓ | ✓ |

*The method is based on the *whitebox* attack, and is extended to *blackbox* by removing a loss term that required the FR model.
**The method is based on the *whitebox* attack, and the *blackbox* attack is performed by knowledge distillation of the FR model.
***The method is based on the *whitebox* attack, and is extended to *blackbox* using a different FR model.

defined two types of attacks and compared the reconstructed face images with the same (type 1) or different (type 2) image of the same subject. However, they did not evaluate the transferability of reconstructed face images.

Duong et al. [2020] and Truong et al. [2022] used a same bijection learning framework and trained a GAN with a generator with structure of PO-GAN [Karras et al., 2017] and TransGAN [Jiang et al., 2021], respectively. While their method is based on the *whitebox* attack, they proposed to use knowledge distillation to extend to the *blackbox* attack. To this end, they trained a student network that mimics the target FR model. However, they did not provide any details (nor source code) about student network training, such as the structure of the student network, etc.

Ahmad et al. [2022] used a GAN-based face reconstruction network to generate low-resolution face images in the *blackbox* scenario. They focus on the size of training dataset and proposed a method to learn the reconstruction network with less training data. While using GAN-based approach, sample reconstructed face images in their paper do not look realistic and suffer from many artifacts.

Dong et al. [2021] used a pre-trained StyleGAN to generate high-resolution face images in the *blackbox* attack against FR systems. They generated synthetic face images using pre-trained StyleGAN and extracted their templates. Then, they trained a fully connected network using mean squared error to map extracted templates to the corresponding noise in the input space $\mathcal{Z}$ of StyleGAN. In contrast, Vendrow and Vendrow [2021], instead of a learning-based approach, used a grid search optimization using the simulated annealing [Van Laarhoven and Aarts, 1987] approach to find the noise $z \in \mathcal{Z}$ in the input of StyleGAN, which generates an image that has the same template. As their iterative method has a large computation cost, they evaluated their method on 20 images only. Along the same lines, Dong et al. [2023] also tried to solve a similar optimization to [Vendrow and Vendrow, 2021] with a different approach. They used the genetic algorithm to find the noise $z \in \mathcal{Z}$ in the input of StyleGAN that can generate an image with the same template.

Compared to most works in the literature that generate low-resolution face images, our proposed method generates high-resolution realistic face images. While low-resolution reconstructed images can be used for evaluating the vulnerability of FR systems under some assumptions, high-resolution images can lead to different types of presentation attacks against FR systems. Previous works in the literature for reconstructing high-resolution face images [Vendrow and Vendrow, 2021, Dong et al., 2021, 2023] tried to find an appropriate noise $z \in \mathcal{Z}$ in the input of StyleGAN that can generate an image with a similar template. However, the input of StyleGAN $z \in \mathcal{Z}$ is a noise and is difficult to control and optimize. In contrast, the intermediate latent space $\mathcal{W}$, which we use in our paper, is more controllable[2]. However, finding appropriate latent code $w \in \mathcal{W}$ in the intermediate space of StyleGAN that can generate an image with a similar facial template is challenging for two reasons. First, for a learning-based approach, we do not have correct values of $w \in \mathcal{W}$ for each real face image to directly use for training the mapping from face templates to the intermediate latent space $\mathcal{W}$ of StyleGAN. Second, as described in Section 2, the mapped latent code should be in the same the intermediate latent space $\mathcal{W}$ of StyleGAN, otherwise the generated image is not face-like.

We should also note that, unlike most works in the literature, we propose our method for both *whitebox* and *blackbox* scenarios and evaluate the transferability of our attack. Similar to [Cole et al., 2017, Duong et al., 2020, Truong et al., 2022], our method is based on the *whitebox* knowledge of FR model, however our approach for extending our method to the *blackbox* attack using another FR model has not been used in TI attacks. Last but not least, we define five different attacks against FR systems and evaluate the vulnerability of SOTA FR models to our attacks. To our knowledge, this is the first paper in which the transferability of reconstructed face images in TI attacks has been investigated.

## 4  Experiments

In this section, we present our experiments and discuss our results. First, in Section 4.1 we describe our experimental setup. Then, we present our experimental results in Section 4.2, and discuss our findings.

---

[2]Several papers in the literature used the intermediate latent space $\mathcal{W}$ of StyleGAN for image editing [Roich et al., 2022, Alaluf et al., 2022, Hu et al., 2022].

### 4.1 Experimental Setup

To evaluate the performance of our method, we consider two SOTA FR models[3], including ArcFace [Deng et al., 2019], ElasticFace [Boutros et al., 2022], as the models from which templates are leaked (i.e., $F_{database}$). For transferability evaluation, we also use three different FR models with SOTA backbones from FaceX-Zoo [Wang et al., 2021], including HRNet [Wang et al., 2020], AttentionNet [Wang et al., 2017], and Swin [Liu et al., 2021], for the target FR system (i.e., $F_{target}$). The recognition performance of these models are reported in Table 3. All these models are trained on

Table 3: Recognition performance of face recognition models used in our experiments in terms of true match rate (TMR) at the thresholds correspond to false match rates (FMRs) of $10^{-2}$ and $10^{-3}$ evaluated on the MO-BIO and LFW datasets. The values are in percentage.

| model | MOBIO | | LFW | |
|---|---|---|---|---|
| | FMR=$10^{-2}$ | FMR=$10^{-3}$ | FMR=$10^{-2}$ | FMR=$10^{-3}$ |
| ArcFace | 100.00 | 99.98 | 97.60 | 96.40 |
| ElasticFace | 100.00 | 100.00 | 96.87 | 94.70 |
| HRNet | 98.98 | 98.23 | 89.30 | 78.43 |
| AttentionNet | 99.71 | 97.73 | 84.27 | 72.77 |
| Swin | 99.75 | 98.98 | 91.70 | 87.83 |

MS-Celeb1M dataset [Guo et al., 2016]. We assume that the adversary does not have access to the FR training dataset, and therefore we use another dataset for training our face reconstruction models. To this end, we use the Flickr-Faces-HQ (FFHQ) dataset [Karras et al., 2019], which consists of 70,000 high-resolution (i.e., $1024 \times 1024$) face images (without identity labels) crawled from the internet. We use 90% random portion of this dataset for training, and the remaining 10% for validation.

To evaluate different attacks against FR systems, we consider two other face image datasets with identity labels, including the MOBIO [McCool et al., 2013] and Labeled Faces in the Wild (LFW) [Huang et al., 2007] datasets. The MOBIO dataset consists of bi-modal (face and voice) data captured using mobile devices from 150 people in 12 sessions (6-11 samples in each session). The LFW dataset includes 13,233 face images of 5,749 people collected from the internet, where 1,680 people have two or more images.

For each of the attacks described in Section 2, we build one or two separate FR systems with one or two SOTA FR models based on the attack type. If the target system is the *same* as the system from which the template is leaked, we have only one FR system. Otherwise, if the target system is *different* the system from which the template is leaked, we have two FR systems with two different feature extractors. In each case, we use one of our evaluation datasets (i.e., MOBIO and LFW) to build both FR systems (so that the subject with the leaked template be enrolled in the target system too). In each evaluation, we assume that the target FR system is configured at the threshold corresponding to a false match rate (FMR) of $10^{-3}$, and we evaluate the adversary's success attack rate (SAR) in entering that system.

We should note that the templates extracted by the aforementioned FR models have 512 dimensions. The input noise $z \in \mathcal{Z}$ to the mapping network of StyleGAN's pre-trained network is from the standard normal distribution and has 512 dimensions. The input noise $n \in \mathcal{N}$ to our mapping network $M_{rec}(.)$ is with dimension of 8 and also from the standard normal distribution. We also use Adam [Kingma and Ba, 2015] optimizer to train our mapping network[4] .

### 4.2 Analysis

In this section, we consider SOTA FR models and evaluate the performance of our face reconstruction method in five different attacks described in Section 2. We also explore the effect of our WGAN training as well as effect of loss terms as our ablation study. In addition, explore important areas in the reconstructed face images that lead to success TI attack. Finally, we discuss limitations of our face reconstruction model.

**Whitebox Knowledge of $F_{database}$** For attacks 1-2, the adversary is assumed to have *whitebox* knowledge of the system from which the template is leaked (i.e., $F_{database}$) and use the same feature extraction model for training (i.e., $F_{loss}$), thus in such cases $F_{loss} = F_{database}$. We considered ArcFace

---

[3]While we use three different face recognition models in our problem formulation, since these models are applied in separate stages, as our experiments in Section A.4 of appendix shows, there is no issue if the inputs and outputs (e.g., pre-processing steps or dimensions) be different in each of these face recognition models.

[4]Source code of our experiment is available at: https://gitlab.idiap.ch/bob/bob.paper.neurips2023_face_ti

Table 4: Evaluation of attacks with *whitebox* knowledge of the system from which the template is leaked (i.e., $F_{\text{loss}} = F_{\text{database}}$) against SOTA FR models in terms of adversary's success attack rate (SAR) using our proposed method on the MOBIO and LFW datasets. The values are in percentage and correspond to the threshold where the target system has FMR $= 10^{-3}$. Cells are color coded according the type of attack as defined in Section 2 for attack 1 ( light gray ) and attack 2 ( dark gray ).

| $F_{\text{database}}$ | MOBIO | | | | | LFW | | | | |
|---|---|---|---|---|---|---|---|---|---|---|
| | ArcFace | ElasticFace | HRNet | AttentionNet | Swin | ArcFace | ElasticFace | HRNet | AttentionNet | Swin |
| **ArcFace** | 92.38 | 81.90 | 71.43 | 70.48 | 74.29 | 86.82 | 74.20 | 36.57 | 36.40 | 58.86 |
| **ElasticFace** | 78.10 | 87.62 | 64.29 | 64.76 | 69.05 | 78.25 | 82.52 | 41.80 | 40.25 | 61.09 |

Table 5: Evaluation of attacks (with *blackbox* knowledge of the system from which the template is leaked i.e., $F_{\text{database}}$) against SOTA FR models in terms of adversary's success attack rate (SAR) using different methods on the MOBIO and LFW datasets. The values are in percentage and correspond to the threshold where the target system has FMR $= 10^{-3}$. **M1**: NbNetB-M [Mai et al., 2018], **M2**: NbNetB-P [Mai et al., 2018], **M3**: [Dong et al., 2021], **M4**: [Vendrow and Vendrow, 2021], and **M5**: [Dong et al., 2023]. Cells are color coded according the type of attack as defined in Section 2 for attack 3 ( lightest gray ), attack 4 ( middle dark gray ), and attack 5 ( darkest gray ).

| $F_{\text{database}}$ | $F_{\text{loss}}$ | $F_{\text{target}}$ | MOBIO | | | | | | LFW | | | | | |
|---|---|---|---|---|---|---|---|---|---|---|---|---|---|---|
| | | | M1 | M2 | M3 | M4 | M5 | Ours | M1 | M2 | M3 | M4 | M5 | Ours |
| **ArcFace** | **ElasticFace** | **ArcFace** | 1.90 | 15.24 | 2.38 | 28.10 | 58.57 | **81.90** | 10.68 | 40.25 | 12.91 | 58.88 | 75.31 | **77.16** |
| | | **ElasticFace** | 1.43 | 11.43 | 4.29 | 15.24 | 37.61 | **73.81** | 8.36 | 34.39 | 6.35 | 29.10 | 50.17 | **68.06** |
| | | **HRNet** | 0.95 | 6.19 | 2.86 | 10.00 | 30.48 | **57.14** | 1.30 | 7.78 | 1.75 | 9.20 | 24.72 | **28.45** |
| | | **AttentionNet** | 0 | 6.67 | 3.33 | 4.29 | 26.67 | **54.29** | 1.33 | 7.17 | 2.29 | 9.17 | 24.16 | **28.87** |
| | | **Swin** | 1.43 | 13.33 | 3.81 | 10.95 | 40.00 | **67.14** | 4.27 | 23.85 | 5.97 | 21.75 | 41.27 | **48.28** |
| **ElasticFace** | **ArcFace** | **ArcFace** | 2.38 | 18.57 | 2.86 | 16.19 | 48.09 | **87.14** | 15.33 | 48.67 | 11.81 | 37.45 | 65.40 | **83.20** |
| | | **ElasticFace** | 3.81 | 43.81 | 4.76 | 43.33 | 72.38 | **89.05** | 21.44 | 58.16 | 11.59 | 52.88 | 74.08 | **83.43** |
| | | **HRNet** | 0.48 | 20.00 | 1.43 | 10.48 | 42.86 | **73.81** | 3.46 | 18.36 | 2.74 | 11.82 | 32.99 | **49.02** |
| | | **AttentionNet** | 1.90 | 18.10 | 3.33 | 9.05 | 40.00 | **71.90** | 2.89 | 16.31 | 2.91 | 10.95 | 31.15 | **46.63** |
| | | **Swin** | 0.95 | 26.19 | 2.86 | 15.24 | 46.67 | **75.24** | 9.22 | 38.79 | 8.26 | 24.62 | 51.20 | **66.89** |

and ElasticFace models and reconstructed face images from the templates extracted by these models in attacks against different FR systems. Table 4 reports the vulnerability of different target systems to our attacks[5] 1-2 in terms of adversary's SAR at the system's FMR of $10^{-3}$. Similar results for the system's FMR of $10^{-2}$ are reported in Table 7 of the appendix. According to these tables, our method achieves considerable SAR against ArcFace and ElasticFace target systems in attack 1. In attack 2, we observe that there is a degradation in SAR with respect to attack 1. However, the reconstructed face images can still be used to enter another target system. Meanwhile, the FR model with a higher recognition accuracy is generally more vulnerable to attack 2. For instance, when ArcFace is considered as $F_{\text{database}}$, we observe that ElasticFace and Swin have the highest SAR as target systems, while there is the same order for their recognition performance in Table 3.

**Blackbox Knowledge of $F_{\text{database}}$** For attacks 3-5, the adversary is assumed to have *blackbox* knowledge of the system from which the template is leaked (i.e., $F_{\text{database}}$) and use another feature extraction model for training (i.e., $F_{\text{loss}}$), therefore in such cases $F_{\text{loss}} \neq F_{\text{database}}$. Table 5 compares the performance of our method with *blackbox* methods[6] in the literature [Mai et al., 2018, Dong et al., 2021, Vendrow and Vendrow, 2021, Dong et al., 2023] for attacks 3-5 in terms of adversary's SAR at system's FMR of $10^{-3}$. Similar results for the FMR of $10^{-2}$ are available in Table 8 of the appendix.

As these tables show, our proposed method achieves the highest SAR compared to [Mai et al., 2018, Dong et al., 2021, Vendrow and Vendrow, 2021, Dong et al., 2023] against FR systems on the MOBIO and LFW datasets. In particular, in attack 5 which is the hardest attack, where $F_{\text{database}}$, $F_{\text{loss}}$, and $F_{\text{target}}$ are different, the results show that the target FR system is still vulnerable to our attack. The results of our method for attack 5 also show transferability of our attack to different FR systems. Similar to attack 2, we can also observe that in attack 5, the FR model with a higher recognition accuracy is generally more vulnerable to our attack.

---

[5]We should highlight that the *whitebox* methods reported in Table 2 do not have available source code, and we could not compare our method with *whitebox* methods in Table 2.

[6]The other *blackbox* methods in the literature do not have available source code and we could not reproduce their results.

Figure 4 also shows sample face images from the LFW dataset and the reconstructed images using our proposed method from ArcFace templates in different attacks. We should highlight that as show in Figure 4, the reconstructed face images in attack 1 and attack 2 are the same, but they are used to enter different target FR system. The same holds for the reconstructed face images in attacks 3-5.

**Ablation Study** To evaluate the effect of WGAN in training our mapping network and the effect of each term in our loss function (i.e., Eq. 1), we consider the ArcFace model in the *white-box* scenario and train different face reconstruction networks with different loss functions. Then, we attack a system with the ArcFace model as a feature extractor (i.e., attack 1) and compare the SARs as reported in Table 6. According to these results, the proposed adversarial training has a significant effect on our face reconstruction method. Because we fix the synthesis network of StyleGAN, the mapped latent codes need to be of the same distribution as $\mathcal{W}$. Otherwise, the generated image is not face-like and training fails to converge. The WGAN training in our method helps our mapping network to learn the distribution of StyleGAN's intermediate latent space, and thus the synthesis network generates

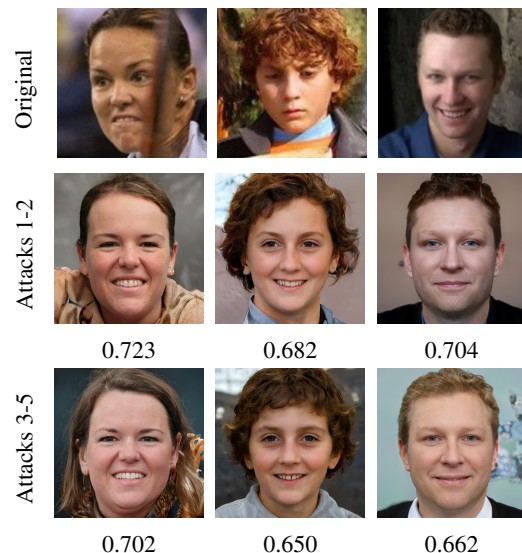

Figure 4: Sample face images from the LFW dataset (first raw) and their corresponding reconstructed images using our template inversion method from ArcFace templates in different attacks, attacks 1-2 (second raw) and attacks 3-5 (second raw, using ElasticFace for $F_{\text{loss}}$). The values below each image show the cosine similarity between the corresponding ArcFace templates of original and reconstructed face images.

face-like images. When we use the WGAN training and based on the results in Table 6, the ID loss has a high impact on the performance of the template inversion model. While the pixel loss by itself does not achieve a good performance, it improves the performance of ID loss in our reconstruction loss function in Eq. 1. This table confirms that the proposed WGAN training and our reconstruction loss function lead to a more successful attack. More experiments for ablation study on the effect of different elements in our proposed method are presented in the appendix.

Table 6: Evaluating the effect of each loss term in our loss function in attack 1 against ArcFace in terms of SAR in the system with FMRs of $10^{-2}$ and $10^{-3}$ evaluated on the MOBIO and LFW datasets. The values are in percentage.

| WGAN training (Eqs. 4 and 5) | Reconstruction Loss Function | MOBIO | | LFW | |
|---|---|---|---|---|---|
| | | FMR=$10^{-2}$ | FMR=$10^{-3}$ | FMR=$10^{-2}$ | FMR=$10^{-3}$ |
| ✓ | $\mathcal{L}_{\text{rec}} = \mathcal{L}_{\text{pixel}} + \mathcal{L}_{\text{ID}}$ | **100.00** | **92.38** | **93.64** | **86.82** |
| | $\mathcal{L}_{\text{rec}} = \mathcal{L}_{\text{ID}}$ | 98.10 | 82.38 | 90.56 | 80.74 |
| | $\mathcal{L}_{\text{rec}} = \mathcal{L}_{\text{pixel}}$ | 0 | 0 | 0.65 | 0.07 |
| ✗ | $\mathcal{L}_{\text{rec}} = \mathcal{L}_{\text{pixel}} + \mathcal{L}_{\text{ID}}$ | 0 | 0 | 0.32 | 0.02 |
| | $\mathcal{L}_{\text{rec}} = \mathcal{L}_{\text{ID}}$ | 0 | 0 | 0.14 | 0.02 |
| | $\mathcal{L}_{\text{rec}} = \mathcal{L}_{\text{pixel}}$ | 0 | 0 | 0.44 | 0.09 |

**Important Areas in the Reconstructed Face Images** As another experiment, we explore important areas in the reconstructed face images. Finding these areas help us to investigate what features between the original template and synthetic images fool the face recognition system, and therefore we can understand what information is encoded in the facial templates. To this end, we apply the Grad-Cam [Selvaraju et al., 2017] algorithm using the face recognition model on the reconstructed face images to see which areas of the reconstructed face images are important and cause the facial templates of our reconstructed face images to be close to the original facial templates. Figure 5 shows results of applying the Grad-Cam algorithm on sample reconstructed face images using our proposed method. As the results in this figure show, important areas that cause the reconstructed face images to have similar templates to the original ones correspond to areas such as eyes, nose, lips, etc. In

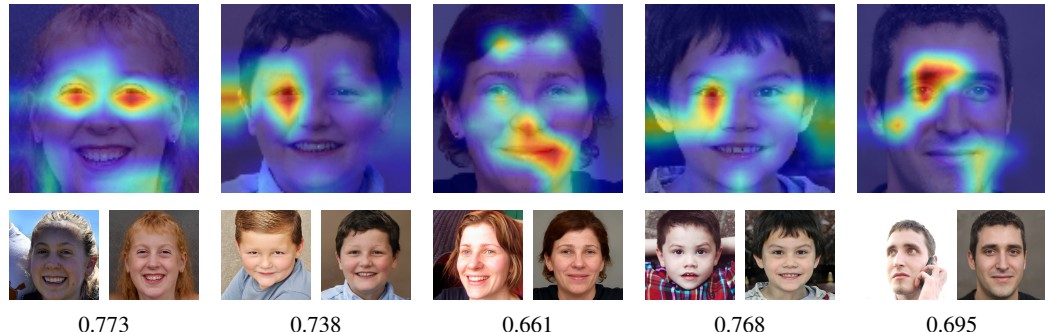

0.773 0.738 0.661 0.768 0.695

Figure 5: Sample face images from the FFHQ dataset and the corresponding important areas in the reconstructed face images using the Grad-Cam algorithm. The reconstructed face images are generated from ArcFace templates. In each example, the top image is the output of the Grid-Cam algorithm, and the bottom images are real (right) and reconstructed (left) images. The value below each sample is the cosine similarity between the templates of original and reconstructed face images.

particular, the area around the eyes seems to be the most important part in most of the reconstructed face images. These results also show that the general shape of the face (e.g., thin or chubby face), hairs, textures, etc., are not often necessarily important in the reconstructed face images, and thus, we can also conclude that these attributes are not well-encoded in the templates extracted by face recognition models.

**Limitations** Despite the significant performance of our method in terms of success attack rate in all types of attacks reported in Table 4 and Table 5, the reconstructed face images fail to enter the system in some cases. Figure 6 illustrates sample failure cases in the attack 3 against ArcFace (using ElasticFace for $F_{loss}$) on the LFW dataset. From the failure cases, we can conclude that there is a bias in the face reconstruction for specific demographics, like elderly or dark skin people. Indeed, such kind of bias in the reconstructed face images is caused by inherent biases in datasets used to train FR model, the StyleGAN model, and our mapping network in our face reconstruction model[7].

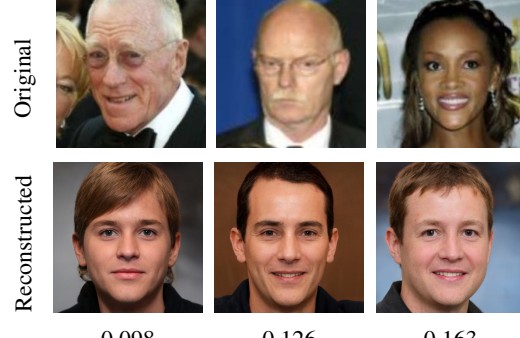

0.098 0.126 0.163

Figure 6: Sample failure cases images from the LFW dataset and their corresponding reconstructed images using our template inversion method from ArcFace templates in the attack 3 (using ElasticFace for $F_{loss}$). The values below each image show the cosine similarity between the corresponding templates of original and reconstructed face images.

## 5 Conclusion

In this paper, we proposed a new method to reconstruct high-resolution realistic face images from facial templates in a FR system. We used a pre-trained StyleGAN3 network and learned a mapping from facial templates to intermediate latent space of StyleGAN within a GAN-based framework. We proposed our method for *whitebox* and *blackbox* scenarios. In the *whitebox* scenario, the adversary can use the feature extraction model for training the face reconstruction network; however, in the *blackbox* scenario, we assume that the adversary has access to another feature extraction model. In addition, we consider the threat model where the adversary might impersonate in the same or another (i.e., transferable attack) FR system. Based on the adversary's knowledge of the feature extraction model and the target FR system, we defined five different attacks and evaluated the vulnerability of SOTA FR systems to our proposed method. Our experiments showed that the reconstructed face images by our proposed method not only can achieve a high SAR in *whitebox* and *blackbox* scenarios, but also are transferable and can be used to enter target FR systems with a different FR model.

---

[7]The biases for different demographies in verification task for ArcFace model are studied in [de Freitas Pereira and Marcel, 2021]. Similarly, biases in StyleGAN generated images and also the FFHQ dataset (i.e., our training dataset) are investigated in [Karakas et al., 2022, Tan et al., 2020, Balakrishnan et al., 2020].

## Acknowledgment

This research is based upon work supported by the H2020 TReSPAsS-ETN Marie Skłodowska-Curie early training network (grant agreement 860813).

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

# A More Analyses

## A.1 Evaluation of *Whitebox* and *Blackbox* Attacks at FMR$= 10^{-2}$

Table 7 and Table 8 of this appendix report the evaluation of attacks with *whitebox* and *blackbox* knowledge, respectively, of the system from which the template is leaked (i.e., $F_{\text{loss}} = F_{\text{database}}$) against SOTA FR models at **FMR$= 10^{-2}$** in terms of adversary's success attack rate (SAR) using our proposed method on the MOBIO and LFW datasets. As the results in these tables show, our method outperforms previous methods in the literature.

Table 7: Evaluation of attacks with *whitebox* knowledge of the system from which the template is leaked (i.e., $F_{\text{loss}} = F_{\text{database}}$) against SOTA FR models in terms of adversary's success attack rate (SAR) using our proposed method on the MOBIO and LFW datasets. The values are in percentage and correspond to the threshold where the target system has **FMR$= 10^{-2}$**. Cells are color coded according the type of attack as defined in Section 2 of the paper for attack 1 ( light gray ) and attack 2 ( dark gray ).

| $F_{\text{database}}$ | MOBIO | | | | | LFW | | | | |
|---|---|---|---|---|---|---|---|---|---|---|
| | ArcFace | ElasticFace | HRNet | AttentionNet | Swin | ArcFace | ElasticFace | HRNet | AttentionNet | Swin |
| **ArcFace** | 100.00 | 93.81 | 80.00 | 81.90 | 85.24 | 93.64 | 90.89 | 68.08 | 62.75 | 76.24 |
| **ElasticFace** | 90.95 | 93.33 | 78.57 | 83.81 | 84.29 | 87.88 | 92.80 | 71.82 | 64.24 | 75.70 |

Table 8: Evaluation of attacks (with *blackbox* knowledge of the system from which the template is leaked i.e., $F_{\text{database}}$) against SOTA FR models in terms of adversary's success attack rate (SAR) using different methods on the MOBIO and LFW datasets. The values are in percentage and correspond to the threshold where the target system has **FMR$= 10^{-2}$**. **M1**: NbNetB-M [Mai et al., 2018], **M2**: NbNetB-P [Mai et al., 2018], **M3**: [Dong et al., 2021], **M4**: [Vendrow and Vendrow, 2021], and **M5**: [Dong et al., 2023]. Cells are color coded according the type of attack as defined in Section 2 of the paper for attack 3 ( lightest gray ), attack 4 ( middle dark gray ), and attack 5 ( darkest gray ).

| $F_{\text{database}}$ | $F_{\text{loss}}$ | $F_{\text{target}}$ | MOBIO | | | | | | LFW | | | | | |
|---|---|---|---|---|---|---|---|---|---|---|---|---|---|---|
| | | | M1 | M2 | M3 | M4 | M5 | Ours | M1 | M2 | M3 | M4 | M5 | Ours |
| **ArcFace** | **ElasticFace** | **ArcFace** | 26.67 | 49.05 | 20.48 | 67.14 | 85.71 | **89.52** | 26.66 | 61.66 | 28.31 | 76.98 | 87.25 | **87.85** |
| | | **ElasticFace** | 11.90 | 49.52 | 16.19 | 34.29 | 60.95 | **86.67** | 32.42 | 66.61 | 23.05 | 57.84 | 74.31 | **87.43** |
| | | **HRNet** | 10.48 | 24.76 | 10.00 | 26.19 | 54.28 | **79.05** | 18.69 | 43.21 | 17.37 | 33.55 | 50.22 | **60.93** |
| | | **AttentionNet** | 11.43 | 38.10 | 18.10 | 24.29 | 54.76 | **80.48** | 10.84 | 31.88 | 13.31 | 26.73 | 44.99 | **53.86** |
| | | **Swin** | 10.48 | 45.24 | 10.95 | 29.52 | 58.09 | **82.86** | 14.79 | 45.80 | 16.98 | 38.03 | 57.71 | **67.80** |
| **ElasticFace** | **ArcFace** | **ArcFace** | 17.14 | 49.05 | 20.95 | 47.14 | 79.91 | **95.24** | 33.08 | 67.89 | 26.35 | 57.48 | 73.80 | **91.23** |
| | | **ElasticFace** | 30.00 | 70.95 | 25.7 | 75.24 | 88.80 | **94.76** | 52.99 | 81.74 | 33.53 | 79.62 | 88.80 | **93.34** |
| | | **HRNet** | 8.10 | 47.14 | 15.24 | 31.43 | 67.14 | **83.81** | 29.27 | 60.34 | 23.22 | 39.06 | 62.01 | **76.68** |
| | | **AttentionNet** | 12.86 | 47.14 | 23.43 | 40.95 | 66.19 | **87.14** | 18.53 | 46.36 | 17.78 | 31.53 | 55.29 | **69.45** |
| | | **Swin** | 10.00 | 54.76 | 13.81 | 37.14 | 68.57 | **89.05** | 24.50 | 60.19 | 21.40 | 41.13 | 65.82 | **80.15** |

## A.2 Ablation Study

**Ablation Study on the Effect of Feature Extractor in the ID loss** To evaluate the effect of feature extractor in our loss function, we consider attack 3 on HRNet templates and use ArcFace and ElasticFace for $F_{\text{loss}}(.)$ in our loss function. Table 9 of this appendix reports the result of this ablation study. Comparing the results of different face recognition models used as $F_{\text{loss}}(.)$ in our loss function, we can see that the mapping which is trained using ArcFace achieves a higher SAR that the mapping that is trained with ElasticFace.

Table 9: Evaluating the effect of ID loss term in our loss function in attack 3 against HRNet in terms of SAR in the system with FMRs of $10^{-2}$ and $10^{-3}$ evaluated on the MOBIO and LFW datasets. The values are in percentage.

| $F_{\text{loss}}$ in ID loss | MOBIO | | LFW | |
|---|---|---|---|---|
| | FMR=$10^{-2}$ | FMR=$10^{-3}$ | FMR=$10^{-2}$ | FMR=$10^{-3}$ |
| ArcFace | **91.90** | **86.19** | **76.01** | **48.22** |
| ElasticFace | 86.71 | 82.38 | 72.59 | 43.71 |

Moreover, comparing these results with the recognition performances of ArcFace and ElasticFace reported in Table 2 of the paper, we can conclude that a face recognition method with a higher recognition performance can lead to a better reconstruction when used as $F_{\text{loss}}$ in the blackbox attack using our proposed method.

**Ablation Study on the Effect of Noise in our WGAN Training**     To evaluate the effect of noise used in our GAN training, we implement another ablation with the same configuration used for our ablation study in the paper (i.e., attack 1 against ArcFace), and we train two networks with and without noise in the input of the mapping network. Table 10 of this appendix reports the result of our ablation study. As this table shows, using noise in our WGAN training improves the performance of our face reconstruction method.

It is noteworthy that generally, in training GANs (even in conditional GANs) a noise (e.g., from Gaussian distribution) is used in the input of the generator network. The samples of noise in the input help the generator to learn the distribution of the output space, and therefore help the generator network to generate outputs from the same distribution of real data. The discriminator (or critic in WGAN) network tries to distinguish if the sample output is from the distribution of real

Table 10: Evaluating the effect of using noise in our method in attack 1 against ArcFace in terms of SAR in the system with FMRs of $10^{-2}$ and $10^{-3}$ evaluated on the MOBIO and LFW datasets. The values are in percentage.

| | MOBIO | | LFW | |
| --- | --- | --- | --- | --- |
| | FMR=$10^{-2}$ | FMR=$10^{-3}$ | FMR=$10^{-2}$ | FMR=$10^{-3}$ |
| with noise | **100.00** | **92.38** | **93.64** | **86.82** |
| without noise | 97.14 | 74.76 | 89.19 | 77.72 |

data or not. In other words, adding random noise in the input makes the training stochastic which is suitable for learning a distribution. In our problem, it is very important that the generated latent code is from the same distribution as the intermediate latent space $\mathcal{W}$ of StyleGAN. In particular, if the generated latent code is not in the same distribution of $\mathcal{W}$, it can easily lead to a non-face-like image at the output of StyleGAN.

**Ablation Study on the Mapping Space**     To evaluate the effect of the mapping space in our proposed method, we consider attack 1 on against ArcFace model, and train mapping to input latent space $\mathcal{Z}$ and the *intermediate* latent space $\mathcal{W}$ of StyleGAN. Table 11 of this appendix reports the result of our ablation study.

As the results in this table show, mapping to the *intermediate* latent space $\mathcal{W}$ leads to a higher performance. This is because the

Table 11: Evaluating the effect of mapping space in our method in attack 1 against ArcFace in terms of SAR in the system with FMRs of $10^{-2}$ and $10^{-3}$ evaluated on the MOBIO and LFW datasets. The values are in percentage.

| Mapping Space | MOBIO | | LFW | |
| --- | --- | --- | --- | --- |
| | FMR=$10^{-2}$ | FMR=$10^{-3}$ | FMR=$10^{-2}$ | FMR=$10^{-3}$ |
| $\mathcal{W}$ | **100.00** | **92.38** | **93.64** | **86.82** |
| $\mathcal{Z}$ | 71.42 | 41.42 | 75.94 | 57.18 |

*intermediate* latent space has more information and is more controllable than input space $\mathcal{Z}$, which is originally of Gaussian distribution for noise in StyleGAN. This ablation study highlights the importance of mapping to the *intermediate* latent space $\mathcal{W}$ of StyleGAN, which has not been proposed in the literature for template inversion.

### A.3   Using a Different Face Generator Network

In our experiments, we used StyleGAN which is one of the most popular face generator models in the literature. However, our method can also be used with other face generator networks. As another experiment, we use StyleSwin [Zhang et al., 2022], which is another face generator model based on transformers. Figure 7 of this appendix shows the reconstructed face images from ArcFace templates using StyleSwin in our method instead of StyleGAN. We used a similar mapping network and learned a mapping from facial templates to the intermediate latent space of StyleSwin. As these results show, our method can also be used with other face generator networks.

### A.4   Application of Our Method for Face Recognition Models with Different Inputs/Outputs

While we use three different face recognition models in our problem formulation, since these models are applied in separate stages, there is no issue if the inputs and outputs (e.g., pre-processing steps or

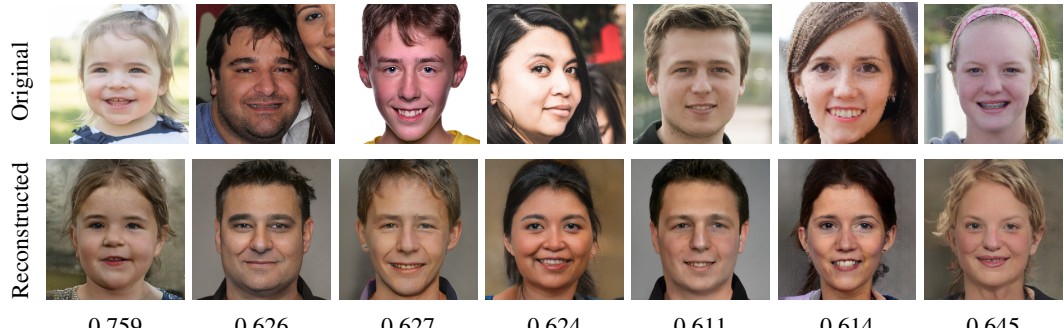

| 0.759 | 0.626 | 0.627 | 0.624 | 0.611 | 0.614 | 0.645 |

Figure 7: Sample face images from the FFHQ dataset and their corresponding reconstructed images from ArcFace templates using our template inversion method with **StyleSwin** [Zhang et al., 2022] as the face generator model. The values below each image show the cosine similarity between the corresponding templates of original and reconstructed face images.

dimensions) be different in each of these face recognition models. For differences in inputs (face images), because each of these models is applied independently on the given face image, the required pre-processing can be considered within the function of the face recognition model in our problem formulation. For differences in outputs (face templates), since the facial templates extracted by each model are compared to facial templates extracted by the same model, there is no conflict in the dimensions. The only point to be noted is that the input of our mapping network should have the same dimension as the templates of .

Let us consider the complete pipeline of our problem formulation as depicted in Figure 2 of the paper. The first face recognition model (i.e., $F_{\text{database}}$) uses its own pre-process and extracts facial templates from face images captured by the camera of the face recognition system (from which the template is leaked). These facial templates (extracted from $F_{\text{database}}$) are then used as input to our face reconstruction model. Therefore, the input of our mapping should have the same dimension as templates of $F_{\text{database}}$. In any case, the output of the face reconstruction network is a high-resolution ($1024 \times 1024$) face image, regardless of the dimension of the input facial template. During training, the generated high-resolution face image is first pre-processed as required by $F_{\text{loss}}$ (i.e., normalised, resized and aligned based on coordinates required by $F_{\text{loss}}$), and the extracted templates are compared with templates of the original image extracted from $F_{\text{loss}}$ (with the required pre-processing for $F_{\text{loss}}$). During inference (i.e., attacking the target FR system), however, the generated high-resolution face image is pre-processed as required by $F_{\text{target}}$. Therefore, there is no conflict in the inputs/outputs in our pipeline.

In our experiments reported in the paper, all face recognition models except Swin take input with $112 \times 112$ resolution. However, the Swin model takes input with $224 \times 224$ resolution. The dimensions of facial templates extracted by all other face recognition models (in Table 3 of the paper) in our experiments are similar and equal to 512. To show that our method can also be used in case of different dimensions of facial templates and to showcase another face recognition model with different pre-processing, as a new experiment, we use a new model, VGGFace [Parkhi et al., 2015], with a different dimension of facial templates (2048-dimension) and different input image resolution ($224 \times 224$) which has a different normalization as well as different

Table 12: Evaluation of success attack rate for TI attack using VGGFace templates (as $F_{\text{database}}$) using ArcFace as $F_{\text{loss}}$ in attack against FR systems with different models (as $F_{\text{target}}$). Note that pre-processing (normalization and alignment coordinates) of VG-GFace is different than all target models and its input resolution is $224 \times 224$. The input resolution for Arc-Face (used as $F_{\text{loss}}$) and ElasticFace is $112 \times 112$ but for Swin is $224 \times 224$. The templates extracted by VGGFace has 2048 dimensions, while templates of ArcFace, ElasticFace, and VGGFace have 512 dimension.

|  | ArcFace | ElasticFace | Swin |
|---|---|---|---|
| FMR $= 10^{-2}$ | 92.92 | 93.10 | 83.97 |
| FMR $= 10^{-3}$ | 86.61 | 82.39 | 72.89 |

landmark coordinates for face alignment. We use ArcFace as our $F_{\text{loss}}$ and evaluate the reconstructed face images in attacks against different face recognition systems (as $F_{\text{target}}$) on the LFW dataset. The results in Table 12 of this appendix show that our proposed method can be applied in the case

where the inputs/outputs of face recognition models in our problem formulation ($F_{\text{database}}$, $F_{\text{loss}}$, and $F_{\text{target}}$) are different, and still achieves high success attack rates against face recognition systems with different inputs/outputs.

## B  Ethics Statement

**Motivations**   The proposed face reconstruction method is presented with the motivation of showing vulnerability of face recognition systems to template inversion attacks. We hope this work encourages researchers of the community to investigate the next generation of safe and robust face recognition systems and to develop new algorithms to protect existing systems. In addition, we should note that the project on which the work has been conducted has passed an Institutional Ethical Review Board (IRB).

**Ethics Considerations**   While the proposed method might pose a social threat against unprotected systems, we do not condone using our work with the intent of attacking a *real* face recognition system or other malicious purposes. We should, however, note that for the next generation of safe face recognition systems, *any kind of potential attacks* should be completely studied by the researchers; and then based upon such vulnerability studies, new protection and defense algorithms will be proposed by the research community in the future. To facilitate future studies, we publish source code of our work as described in Section C of this appendix.

**Mitigation of such Attacks**   This paper demonstrates an important privacy and security threat to the state-of-the-art unprotected face recognition systems. Along the same lines, data protection frameworks, such as the European Union General Data Protection Regulation (EU-GDPR) [European Council, 2016], put legal obligations to protect biometric data as sensitive information. To this end and to prevent such attacks to face recognition systems, several biometric template protection algorithms are proposed in the literature [Nandakumar and Jain, 2015, Sandhya and Prasad, 2017, Kaur et al., 2022, Kumar et al., 2020, Shahreza et al., 2022, 2023].

## C  Reproducibility Statement

In our experiments, we use PyTorch package and the pre-trained model of StyleGAN3[8] and StyleSwin[9] to generate high-resolution face images. We train our mapping network for 16 epochs with an initial learning rate of $0.1$ using Adam optimizer [Kingma and Ba, 2015] and divide the learning rate by 2 every three epochs. Training our mapping network using our proposed method takes around two days on a system equipped with an NVIDIA GeForce RTX[TM] 3090. We build face recognition pipelines using Bob [Anjos et al., 2012, 2017] toolbox[10]. The source code of our experiments is publicly available[11] to help reproduce our results.

## D  Licenses and Copyright Permissions

**Datasets**   We have signed the licenses (GDPR compliance) to use from the data controller of any of the datasets used in this paper (i.e., MOBIO, LFW, and FFHQ) and followed the terms of use of these datasets in this paper. We have also cited the corresponding paper for each dataset.

**Models**   We used pretrained models of following deep neural networks and followed the license of each one in implementing our experiments:

- ArcFace, ElasticFace, and VGGFace face recognition models implemented in Bob [Anjos et al., 2012, 2017] toolbox (under BSD 3-Clause License)
- HRNet, AttentionNet, and Swin face recognition models implemented in FaceX-Zoo [Wang et al., 2021] toolbox (under Apache License, Version 2.0)

---

[8] Available at `https://github.com/NVlabs/stylegan3`

[9] Available at `https://github.com/microsoft/StyleSwin`

[10] Available at `https://www.idiap.ch/software/bob/`

[11] Available at `https://gitlab.idiap.ch/bob/bob.paper.neurips2023_face_ti`

- StyleGAN3 (official) model published under Nvidia Source Code License[12].
- StyleSwin (official) model published under MIT License.

---

[12] Available at `https://github.com/NVlabs/stylegan3/blob/main/LICENSE.txt`

