# OpenReview forum: "Face Reconstruction from Facial Templates by Learning Latent Space of a Generator Network"
_NeurIPS.cc/2023/Conference — NeurIPS 2023 poster_

### Official Review · Reviewer_dWCe · 2023-06-28

**Soundness:** 3 good
**Presentation:** 2 fair
**Contribution:** 2 fair
**Rating:** 5
**Confidence:** 4

**Summary:**

The paper presents a method to reconstruct high resolution face images from feature vectors extracted from a face recognition (FR) system. The reconstructed face images can be used to attack FR systems to gain access under whitebox and blackbox scenarios
The paper also introduce five different scenarios that one can use to attack FR systems. The authors raised awareness to protect template feature vectors stored in FR systems' databases to avoid possible adversary attacks.

**Strengths:**

The paper evaluates various attack scenarios with five different face recognition models.
The reconstructed face images have high quality. Some of them have similar IDs with the original images.

**Weaknesses:**

1. The novelty in the proposed framework is limited. Most of the components are based on the existing works, e.g. StyleGAN3 and WGAN.
2. The proposed attack scenarios are straightforward as a combination of three possible feature extraction models. The authors should also rate the practical of the scenarios in addition to the level of difficulty since some scenarios cannot achieve in practice.
3. In their problem formulation, the authors consider three different feature extraction models. However, there is an implicit assumption that the input and output of those models are the same. The authors should analyze cases when the input and output are different, e.g. handling various output dimension, various pre-processing steps.
4. There are some errors in referring to tables in the paper, e.g. line 281, 287 and 300.

**Questions:**

See above weakness.

**Limitations:**

The authors addressed one limitation of the reconstruction model. However, the authors should address some other limitation mentioned in the weakness section above.

---

> ### Author Rebuttal · Authors · 2023-08-09
>
> We thank the reviewer for their time in reviewing our paper and for the comments.
> Below, we tried to address the concerns raised by the reviewer:
>
> > The novelty in the proposed framework is limited. Most of the components are based on the existing works, e.g. StyleGAN3 and WGAN.
>
> We acknowledge the reviewer's comment that our method leverages WGAN training to learn a mapping network that maps the facial templates to the intermediate latent space of StyleGAN. However, as shown in our experiments (especially our ablation study), training such a mapping network is not straightforward, and each part of our proposed method contributes to the performance of our method.
> Therefore, our proposed face reconstruction method is not a mere trivial combination of existing techniques. Even more, the proposed method achieves state-of-the-art results in template inversion attacks against five state-of-the-art face recognition systems on different face recognition datasets, which is not trivial either.
>
> > The proposed attack scenarios are straightforward as a combination of three possible feature extraction models. The authors should also rate the practical of the scenarios in addition to the level of difficulty since some scenarios cannot achieve in practice.
>
> Following the reviewer's suggestion, we prepared a table in our `general response` and further described the adversary's knowledge and difficulty of each attack scenario. We will include this table in the final version of the paper. We would like to highlight that we define five different attacks against face recognition systems (based on the adversary's knowledge and the target system). In particular, we evaluate the transferability of the reconstructed face images and the vulnerability of SOTA face recognition models to template inversion attacks, which have not been investigated in the literature. To our knowledge, this is the first work that comprehensively evaluates the transferability of the reconstructed face images in template inversion attacks.
>
> > In their problem formulation, the authors consider three different feature extraction models. However, there is an implicit assumption that the input and output of those models are the same. The authors should analyze cases when the input and output are different, e.g. handling various output dimension, various pre-processing steps.
>
> While we use three different face recognition models in our problem formulation, there is no issue if the pre-processing steps or dimensions be different in each of these face recognition models.
> For differences in inputs (face images), because each of these models is performing independently on the given face image, the required pre-processing can be considered in the function of the face recognition model in our problem formulation. For differences in outputs (face templates), since the facial templates extracted by each model are compared to facial templates extracted by the same model, there is no conflict in the dimensions. The only point to be noted is that the input of our mapping network should have the same dimension as the templates of $F_{databse}$.
>
> Let us consider the complete pipeline of our problem formulation as depicted in Figure 2 of the paper. The first face recognition model $F_\{database}$ uses its own pre-process and extracts facial templates from face images captured by the camera of face recognition system. These facial templates (extracted from $F_\{database}$) are then used as input to our face reconstruction model. Therefore, the input of our mapping should have the same dimension as templates of $F_\{database}$. In any case, the output of the face reconstruction network is a high-resolution face image. During training, this high-resolution face image is first pre-processed as required by $F_\{loss}$ and the extracted templates are compared with templates of the original image extracted from $F_\{loss}$. During inference, the generated high-resolution face image is pre-processed as required by $F_\{target}$. Therefore, there is no conflict in the input/outputs in our pipeline.
>
> In our experiments reported in the paper, all face recognition models except Swin take input with  $112\times112$ resolution. However, the Swin model takes input with  $224\times224$ resolution.
> However, we acknowledge that the dimensions of facial templates extracted by all face recognition models in our experiments are similar and equal to 512.
> To show that our method can be used in case of different dimensions of facial templates and to showcase another face recognition model with different pre-processing, as a new experiment, we used a new model (VGGFace) with **different dimension of facial templates** (2048-dimension) and **different input image resolution** ($224\times224$) which has a **different normalisation** as well as **different landmark coordinates for face alignment**. We used ArcFace as our $F_\text{loss}$ and evaluated the reconstructed face images in attacks against different face recognition systems (as $F_{target}$) on the LFW dataset:
>
> |  | ArcFace | ElasticFace| Swin |
> |---|---|---|---|
> | $\text{FMR}=10^{-2}\%$ | 92.92 | 93.10 | 83.97 |
> | $\text{FMR}=10^{-2}\%$ | 86.61 | 82.39 | 72.89 |
>
> As the results in this table show, our proposed method can be applied in the case that the input/output of face recognition models in our problem formulation ($F_{database}$, $F_{loss}$, $F_{target}$) are different.
>
> > There are some errors in referring to tables in the paper, e.g. line 281, 287 and 300.
>
> We apologize for this error caused in our compilation before submission. We will fix these errors and will carefully revise the paper for any typos/errors.
>
>
> > Flag For Ethics Review
>
> We have already included a section in the supplementary material and discussed different ethical aspects.
> Regarding the used datasets, we have also included a section in our supplementary material and discussed the licenses.

---

> > ### Comment · Reviewer_dWCe · 2023-08-10
> >
> > Thank you for your rebuttal. I have read it and will respond with points for further discussion.

---

> > > ### Author Response · Authors · 2023-08-11
> > > **Reply by Authors**
> > >
> > > We thank the reviewer for their time in reading our rebuttal.
> > >
> > > We would like to also mention that in addition to our individual rebuttal to answer the reviewer's question; we have also conducted new experiments as described in the `general response` based on comments we received from reviewers and would like to *add these new experiments in our final version* to improve the quality of the paper.
> > >
> > > We are more than happy to continue our discussion with the respected reviewer and to address any remaining/new concerns.

---

> > > > ### Comment · Reviewer_dWCe · 2023-08-21
> > > >
> > > > Thank you for clarifying all my concerns. My concerns were addressed in the authors' response. I will raise my initial rating.
> > > > It would be great to include the analysis that the proposed face reconstruction method is not a mere trivial combination of existing techniques as well as the table and experiments as promised.

---

> > > > > ### Author Response · Authors · 2023-08-21
> > > > > **Reply by Authors**
> > > > >
> > > > > We thank the reviewer for their feedback and also for raising their initial rating.
> > > > > We are happy that our reply could clarify and address all reviewer's concerns.
> > > > > In the final version, we will elaborate on the points mentioned by the reviewer, and also we will include the table and experiments which we provided in our rebuttal.

---

> > > ### Comment · Senior_Area_Chairs · 2023-08-21
> > > **From AC & SAC: Please respond with points by August 21**
> > >
> > > Hi Reviewer dWCe,
> > >
> > > As commented in your previous post, "I have read it and will respond with points for further discussion." So through these days, we are anticipating your further feedback, which is crucial for making a fair and transparent decision.
> > >
> > > **Please help us move forward by entering your feedback by August 21.** Your support is crucial to guarantee the review quality of NeurIPS, and is deserving our warm thanks.
> > >
> > > Best,
> > > AC & SAC

---

### Official Review · Reviewer_pmB2 · 2023-06-28

**Soundness:** 3 good
**Presentation:** 2 fair
**Contribution:** 3 good
**Rating:** 5
**Confidence:** 4

**Summary:**

The paper studies the template inversion attack on FR systems. The paper involves both white- and black-box attacks. Moreover, five different attacks are considered.

**Strengths:**

This paper poses a potential threat to FR systems in that attackers can reconstruct victim's facial images with the features stored in the database. Comprehensive attack situations are considered with five different attacks.

**Weaknesses:**

- My main concern is about the limited application of the proposed method. The quality of the face images highly depends on the power of the generator, i.e. StyleGAN3. Therefore, the bias of it, such as resolution, gender, race, and head pose biases make the attack hardly generalize to other face images. For example, including the failure Fig. 5, if the features stored in the database are from low-resolution images, or non-frontal faces, or faces aligned with other alignment methods, the proposed method may also fail as these also are the biases/domain gap of the generator. It would be good if the authors analyzed more in this aspect that there is a large domain gap between the database and that in the generator. From the current setting, I have not seen any specific handle for this issue but a simple critic network.

**Questions:**

See the Weakness.

**Limitations:**

See the Weakness.

---

> ### Author Rebuttal · Authors · 2023-08-09
>
> We thank the reviewer for their valuable comments. We are happy that the reviewer found our paper easy to follow. We appreciate the reviewer's comments on the strengths of our work. Below, we tried to address the concerns raised by the reviewer:
>
> > My main concern is about the limited application of the proposed method. The quality of the face images highly depends on the power of the generator, i.e. StyleGAN3. Therefore, the bias of it, such as resolution, gender, race, and head pose biases make the attack hardly generalize to other face images. For example, including the failure Fig. 5, if the features stored in the database are from low-resolution images, or non-frontal faces, or faces aligned with other alignment methods, the proposed method may also fail as these also are the biases/domain gap of the generator. It would be good if the authors analyzed more in this aspect that there is a large domain gap between the database and that in the generator. From the current setting, I have not seen any specific handle for this issue but a simple critic network.
>
> In our experiments, we tried to consider the gap between our training dataset and our evaluation dataset. For training our face reconstruction network, we used the FFHQ dataset, which includes high-quality images. In contrast, for evaluation, we used MOBIO and LFW datasets. The MOBIO dataset is collected with mobile devices, and LFW is an unconstrained dataset collected from the internet. Therefore, the quality of images in these two datasets is considerably different from the FFHQ dataset, which is used for our training.
> As a new experiment, we consider the IARPA Janus Benchmark C (IJB-C) dataset, which is one of the most challenging face recognition benchmarking datasets. The following table compares the performance of our method with previous methods in the literature (compared in Table 4 of the paper) in attack 3 (i.e., blackbox attack against same system) using ArcFace as $F_{loss}$ against different state-of-the-art face recognition models on the IJB-C dataset:
>
> |  | M1 | M2 | M3 | M4 | M5 | Ours |
> |---|---|---|---|---|---|---|
> | ElasticFace | 0.32 | 4.1 | 0.13 | 5.41 | 16.90 | **35.91** |
> | HRNet | 0.05 | 1.12 | 0.09 | 2.17 | 4.36 | **24.38** |
> | AttentionNet | 0.13 | 1.27 | 0.21 | 2.86 | 4.82 | **26.35** |
> | Swin | 2.40 | 15.11 | 2.45 | 20.73 | 30.91 | **45.00** |
>
> (Note: M1, M2, M3, M4, and M5 are defined in the caption of Table 4 of the paper.)
>
> As the results in this table show, our method still achieve superior performance than all face reconstruction methods in the literature. We should also note that we observe a drop in the performance of all methods in this table. It is particularly because the IJB-C dataset is a very difficult benchmarking dataset. To elaborate on this, we would like to compare the recognition performance of ArcFace (as an example) face recognition model on IJB-C with the performance on MOBIO and LFW datasets as reported in the following table:
>
> |  | MOBIO | LFW | IJB-C |
> |---|---|---|---|
> | ${FMR}=10^{-2}\%$ | 100.00 | 97.60 | 95.29 |
> | ${FMR}=10^{-3}\%$ | 99.98 | 96.40 | 90.90 |
>
> As the values in this table show, IJB-C is a more challenging dataset, and the face recognition systems also suffer from the degradation in the recognition performance on this dataset.
>
> > Flag For Ethics Review: Ethics review needed: Inappropriate Potential Applications & Impact  (e.g., human rights concerns)
>
> We have already included a section in the supplementary material for the "Ethics Statement" and discussed different ethical aspects of our paper.

---

> > ### Comment · Reviewer_pmB2 · 2023-08-11
> >
> > The rebuttal mainly answered my questions and I keep my scores.

---

> > > ### Author Response · Authors · 2023-08-11
> > > **Reply by Authors**
> > >
> > > We thank the reviewer for reading our rebuttal and are happy that our reply could mainly answer the reviewer's questions.
> > >
> > > We would like to add to our previous response that the quality of facial images mainly affects the feature extraction (i.e., face recognition model), which indirectly affects the performance of our face reconstruction network. Therefore, degradation in the performance of face reconstruction methods that take facial templates as input is inevitable for templates extracted from low-quality face images, while such facial templates also degrade the recognition accuracy of the face recognition system.
> > >
> > > Regarding bias in the reconstructed face images for different demographics (such as age and ethnicity), we have already discussed in the "Limitations" section of our paper that such results are caused by inherent biases in datasets used to train the face recognition model, the face generator model (StyleGAN), and our mapping network. Indeed training the models, particularly the face generator model and the mapping network, with a balanced dataset can help mitigate such biases.
> > >
> > > Regarding the reviewer's comment on the alignment of the face images with different landmark coordinates, we would also like to mention that we implemented a new experiment in reply to R5 (Reviewer dWCe) and considered a challenging scenario where the pre-processing and alignment of face images for $F_{database}$, as well as the size of facial templates, are different than $F_{loss}$ and $F_{target}$.
> > > Our experiment shows that our method can still achieve high success attack rates in attacks against systems with different pre-processing and different input (face image) and output (facial template) dimensions. We ask the reviewer to kindly check these results and our description in our reply to  R5 (Reviewer dWCe).
> > >
> > > We would like to also mention that we have also conducted new experiments as described in the `general response` based on comments we received from other reviewers and would like to **add these new experiments in our final version**, to improve the quality of the paper. We would like to ask the reviewer to also kindly read our new experiments described in the `general response` and if having new experiments can help increase the reviewer's scores.
> > >
> > > In case our current reply or our new experiments described in the general response and in reply to R5 caused any new doubts or questions, we would be more than happy to continue the discussion with the respected reviewer.

---

### Official Review · Reviewer_RLF3 · 2023-07-06

**Soundness:** 3 good
**Presentation:** 3 good
**Contribution:** 3 good
**Rating:** 5
**Confidence:** 4

**Summary:**

In this paper, the authors introduce a new method for reconstructing high-resolution realistic face images from facial templates within a face recognition (FR) system. They employ a pre-trained StyleGAN3 network and train a mapping from facial templates to the intermediate latent space of StyleGAN using a GAN-based framework. In particular, the proposed method is designed to work in both white-box and black-box under different adversary attack scenarios.  The vulnerability of state-of-the-art (SOTA) FR systems to the proposed method is evaluated. Experimental results demonstrate that the reconstructed face images using the proposed method achieves the highest success rates in both white-box and black-box scenarios and its transferability has been validated.

**Strengths:**

+ Overall the paper is well organized and easy to read.
+ The idea of training a critic network to ensure the same distribution between $w \in W$ and ${\hat w} \in W$ is interesting.
+ Deep dive into five different template inversion (TI) attacks, and evaluate the transferability of reconstructed face images in TI attacks.
+ Solid experiment validation based on StyleGAN with better performance than the SOTA methods.


**Weaknesses:**

- The investigation on face reconstruction is insufficient. The authors didn’t mention any work related to Transformer and diffusion works, e.g, Face-Transformer (2023 Arxiv April), and VQ-DDM (CVPR’22).

- The template inversion (TI) attack is a too specific adversarial attack. I would like to suggest the authors experimentally verify whether it is possible to extend the idea (e.g., critic network) to other kinds of attacks.

- The current paper is heavily dependent on the StyleGAN3, which is not SOTA when considering the more advanced Transformer and diffusion models. Therefore I would like to suggest the authors to switch the current StyleGAN3 to the both Transformer and diffusion models to validate the claims made in the current paper.

- Typo: miss the table number in line 287, 300.


**Questions:**

NA.

**Limitations:**

As mentioned by the authors, there exists a bias in reconstructing faces of certain demographic groups, such as elderly individuals or people with dark skin. This bias in the reconstructed face images can be attributed to the inherent biases present in the datasets used to train the face recognition (FR) model, the StyleGAN model, and the mapping network in our face reconstruction model.

---

> ### Author Rebuttal · Authors · 2023-08-09
>
> We thank the reviewer for their valuable comments. We are happy that the reviewer found our paper well-organized and easy to read. We appreciate the reviewer's comments on the strengths of our work.
> Below, we tried to address the concerns raised by the reviewer:
>
>
> > The investigation on face reconstruction is insufficient. The authors didn’t mention any work related to Transformer and diffusion works, e.g, Face-Transformer (2023 Arxiv April), and VQ-DDM (CVPR’22).
>
> We used StyleGAN since it is one of the most popular face generator models in the literature.
> However, our method can also be used with other face generator networks. As a new experiment, we used StyleSwin (CVPR, 2022), which is another face generator model based on transformers. As the results (reported in our `general response`) show, our method can also be used with this face generator networks too.
>
> We should note that considering the NeurIPS guideline, papers appearing less than two months before the submission deadline are considered contemporaneous to NeurIPS submissions. Therefore, Face-Transformer (2023 Arxiv April), which was published on arxiv less than one month before NeurIPS submissions, is considered contemporaneous following the NeurIPS guidelines. In addition, according to the NeurIPS guideline, authors are also excused for not knowing about works published on arxiv.
> In any case, our new experiment shows that our method can be used with recent face generator models based on transformers too.
>
> We would like to also highlight that we have considered five different face recognition models in our experiments. One of the face recognition models in our experiments is based on the Swin backbone, which is a transformer-based network. Our experiment shows that the Swin face recognition model is also vulnerable to our attacks.
>
> > The template inversion (TI) attack is a too specific adversarial attack. I would like to suggest the authors experimentally verify whether it is possible to extend the idea (e.g., critic network) to other kinds of attacks.
>
> We appreciate the reviewer's comment and will consider it as our future work. In fact, our method can be extended to any type of attack in which we would like to find/modify the intermediate latent codes but ensure that the new latent codes are on the original distribution of the intermediate latent space.
>
> > The current paper is heavily dependent on the StyleGAN3, which is not SOTA when considering the more advanced Transformer and diffusion models. Therefore I would like to suggest the authors to switch the current StyleGAN3 to the both Transformer and diffusion models to validate the claims made in the current paper.
>
> As described in our reply to the earlier point raised by the reviewer and following the reviewer's suggestion, as a new experiment, we used StyleSwin (CVPR, 2022), which is another face generator model based on transformers. The results reported in the `general response` show that our method can also be used with this face generator network too. We will add these results to the final version of the paper.
>
> We should not that since our proposed method maps to the intermediate latent space of face generator model, it can be applied to different face generator models. During rebuttal period, we could investigate and showcase the application of our method with StyleSwin (which is a state-of-the-art face generator model based on transformers). Further experiments with other face generator networks can be explored in future works.
>
>
> > Typo: miss the table number in line 287, 300.
>
> We apologize for this error caused in our compilation before submission. We will fix these errors and will carefully revise the paper for any typos/errors.

---

### Official Review · Reviewer_T7AV · 2023-07-08

**Soundness:** 3 good
**Presentation:** 3 good
**Contribution:** 2 fair
**Rating:** 4
**Confidence:** 4

**Summary:**

The paper proposes a high-resolution face reconstruction method for template inversion attacks. They make use of a GAN-based framework, StyleGAN3, by learning a mapping from facial templates to its intermediate latent space. They evaluate their method on 5 different attacks in whitebox and blackbox scenarios.

**Strengths:**

The method is evaluated in a wide range of TI attack scenarios.

The method, its motivation and evaluation setting is described clearly and is easy to understand.


**Weaknesses:**

The novelty of the proposed method mainly lies in the introduction of the facial template mapping network which is somewhat incremental.

It would be helpful to understand the rationale behind selecting StyleGAN as the synthesis model. Additionally, exploring alternative synthesis networks could provide insights into how much of the success and limitations depend on the synthesis network. Also, it would be beneficial to explore the effect of fine-tuning the synthesis network of StyleGAN.

**Questions:**

Please see the weakness.

**Limitations:**

Yes, the limitations are addressed.

---

> ### Author Rebuttal · Authors · 2023-08-09
>
> We thank the reviewer for their time in reviewing our paper and for their valuable comments. We are happy that the reviewer found our paper clear and easy to understand. Below, we tried to address the concerns raised by the reviewer:
>
> > The novelty of the proposed method mainly lies in the introduction of the facial template mapping network which is somewhat incremental.
>
> We acknowledge the reviewer's comment that we eventually aim to train one mapping network in our proposed method to map the facial templates to the intermediate latent space of StyleGAN. However, as shown in our experiments (especially our ablation study), training such a mapping network is not straightforward. In particular, using a WGAN learning approach, we simultaneously train a critic network to help our mapping network to learn the distribution of the intermediate latent space $\mathcal{W}$ of StyleGAN.  In addition to adversarial training in our proposed method to learn the mapping, we also train our mapping network with a multi-term loss function, including an identity loss on the generated images to preserve the identity of the reconstructed face images. Therefore, our proposed face reconstruction method is not a mere trivial combination of existing techniques. Even more, the proposed method achieves state-of-the-art results in template inversion attacks against five state-of-the-art face recognition systems on different face recognition datasets, which is not trivial either.
>
> Moreover, we define five different attacks against face recognition systems (based on the adversary's knowledge and the target system), and evaluate the transferability of the reconstructed face images and vulnerability of state-of-the-art face recognition models to template inversion attacks. To our knowledge, this is the first work that comprehensively evaluates the transferability of the reconstructed face images in template inversion attacks.
>
> > It would be helpful to understand the rationale behind selecting StyleGAN as the synthesis model. Additionally, exploring alternative synthesis networks could provide insights into how much of the success and limitations depend on the synthesis network.
>
> StyleGAN is one of the most popular face generator models in the literature. It has available source code (and pretrained models), and several works have been using StyleGAN in different research problems. However, our method can also be used with other face generator networks. As a new experiment, we used StyleSwin (CVPR, 2022), which is another face generator model. As the results (reported in the `general response`) show, our method can be used with this face generator networks too.
>
> > Also, it would be beneficial to explore the effect of fine-tuning the synthesis network of StyleGAN.
>
> We fix the parameters of the synthesis network of the face generator network (StyleGAN) and do not update it during training. If we update the synthesis network of SyleGAN, we need to apply adversarial training on the generated images too, which makes the training more complicated. That means we need to learn again the distribution of real images, which is more difficult than learning the distribution of intermediate latent space of pretrained StyleGAN.

---

> ### Comment · Senior_Area_Chairs · 2023-08-21
> **From AC & SAC: Please respond by August 21**
>
> Hi Reviewer T7AV,
>
> As all fellow reviewers have seen, your original review was too short, whereas authors have provided a rebuttal to address your concerns. Therefore, it is super important if you can help out by providing some meaningful feedback before August 21, which will help us move forward to a smooth decision.
>
> Your support is highly appreciated.
> Best,
> AC & SAC

---

### Official Review · Reviewer_S2tQ · 2023-07-10

**Soundness:** 2 fair
**Presentation:** 3 good
**Contribution:** 2 fair
**Rating:** 5
**Confidence:** 5

**Summary:**

In this work, the authors propose a new method to execute the template inversion attack against face recognition systems. By simply having access to the feature vectors (latent representations) of the original faces, the adversary can reconstruct a face like the original, therefore showing that face recognition systems can be compromised. The method used comprises a StyleGAN3 network, split such that a new Mapping network attempts to synthesize a correct embedding given random noise and the victim template vector, and a critic network discriminates between that output and StyleGAN’s own mapping network with the WGAN algorithm.
The authors further test their method in whitebox and blackbox scenarios, using a combination of several models for template vector leaking and transferability of reconstructing a face to attack a face recognition system. In the end, for the five attacks defined, the novel method achieves high attack success rates.

**Strengths:**

While the problem is not new, the paper takes a clever spin on existing methods and combines them to tackle five attack scenarios. Even in the most difficult attack scenario, the novel method achieves significantly higher attack success rates, but even more so, the better recognition models actually suffer the most.

It is interesting and reasonable to propose face template matching via the GAN-base framework.

The paper provides sufficient evaluation results for five attack scenarios, as the paper defined.

**Weaknesses:**

The paper contains grammatical errors throughout, but they should be fixable. For instance, in line 108, I believe you meant to say “critique” the generated omega-hat vectors. For Section 4.2, I believe it should be titled “Analysis”.

The five attack scenarios as defined in the paper seem incremental so they do not show the broad of the method application.

The synthesized images themselves in the sense that the authors are not trying to answer the question of *what* features between the template and synthetic images fool the recognition system. I think having that aspect of the attack would both boost the novelty of the attack and provide an excellent visualization as to why (and literally where) the attack works.

**Questions:**

1. Is there a particular reason why a model like StyleGAN3 was chosen? (Code availability, SOTA, etc.).
2. (Related to Weakness section) Is it reasonable to think that the template vectors would be stored “raw” and not in an encrypted form? I do not think a database concerned with security would store the raw information.
3. It seems in many, if not all, images shown, the face subjects have smooth faces (this is somewhat related to elderly subjects reconstructing poor results). Do facial features (e.g., wrinkles, pores, moles, etc.) have any influence on the system recognition succeeding or failing? Perhaps this could be attributed to StyleGAN synthesis of images.
4. The paper should provide an insightful comparison of the attack scenarios, i.e. what scenario A has but B does not have, and vice versa. It looks like the evaluation of the blackbox scenario is enough since it covers all of the others.

**Limitations:**

The authors have addressed the limitations of the work and its biases and have stated which data they have used for experiments (and how the data influences the outcomes presented).

---

> ### Author Rebuttal · Authors · 2023-08-09
>
> We thank the reviewer for their valuable comments. Below, we tried to address the concerns raised by the reviewer:
>
> ### Reply to Weaknesses
> **Reply to Weakness 1:** We acknowledge the reviewer's comment and apologize for the typos as well as grammatical errors. We will fix these errors and will carefully revise the paper for any typos/errors.
>
> **Reply to Weakness 2:** We define five different attacks against face recognition systems (based on the adversary's knowledge and the target system) to provide a comprehensive vulnerability evaluation. In particular, we evaluate the transferability of the reconstructed face images in template inversion attacks, which has not been investigated in the literature.
>
> **Reply to Weakness 3:**
> We appreciate the reviewer for valuable comment and interesting suggestion. We conducted a new experiment to gain a more in-depth insight of the reconstructed face images. Please kindly check our new experiment described in the `general response`.
>
> ***
> ### Reply to Questions
> **Reply to Questions 1:**
> StyleGAN is one of the most popular face generator models in the literature. It has available source code (and pretrained models) and several works have been using StyleGAN in different research problems.
> However, our method can also be used with other face generator networks. As a new experiment, we used StyleSwin (CVPR, 2022) which is another face generator model. As the results (reported in the `general response`) show, our method can be used with this face generator networks too.
>
> **Reply to Questions 2:**
> Data protection regulations consider biometric data as sensitive information that should be protected. However, many such regulations are issued recently and there are still many face recognition systems which store raw facial templates in their databases. We should note that using typical encryption schemes (such as hashing) cannot be used to protect facial templates. Because biometric templates of the same subject are not exactly the same due to variations in measurements (eg light change, pose change, etc), and thus exact matching (as used in hashing) cannot be used in practice for face recognition systems. Hence, protection of biometric templates is still a challenging problem and there are several standards (eg, ISO/IEC 24745) defining the requirements of template protection schemes. In our Ethics Statement, we cited some works on biometric template protection in the literature.
>
> **Reply to Questions 3:**
> To answer the question of `Do facial features have any influence on the system recognition succeeding or failing [of the reconstructed face images]?`, we first need to answer the question that `Do face recognition models encode such facial features?` or in other words  `Do facial features (e.g., wrinkles etc) are useful for face recognition models to identify different subjects?`.
> To have a solid answer to these questions, we need to have access to a rich dataset of facial features and investigate the effect of facial features on the performance of face recognition models and our face reconstruction method. Unfortunately, we could not find any large-scale dataset with labels of such facial features. However, we would like to consider aging as a special case of causing facial features in people and investigate the effect of aging on the performance of face recognition models and our face reconstruction method. For instance, wrinkles often appear more (or strengthen) in elderly people, while the same person does not have (or has less) wrinkles in his/her younger images (as shown in Figure 3 of our rebuttal pdf file).
> To this end, we consider the AgeDB dataset, which contains 16,488 images of various famous people (a total of 568 distinct
> subjects). Every image is annotated with respect to the identity and includes age attributes.
> The minimum and maximum age is 1 and 101, respectively, and the average age range for each subject is 50.3 years.
> This dataset has four different protocols, where in each protocol, the age difference of each pair’s faces is equal to a fixed, predefined value, i.e., 5, 10, 20 and 30 years. As a new experiment, we consider attack 1 (ie, whitebox against the same model) against ArcFace and evaluate the performance of the face recognition model (in terms of true match rate)  and our face reconstruction method (in terms of success attack rate) for different age protocols in this dataset (at FMR=1%):
>
> |  | AgeDB-5 | AgeDB-10 |  AgeDB-20 | AgeDB-30 |
> |---|---|---|---|---|
> | Face recognition (TMR) | 98.33 | 98.43 | 97.30 | 97.00 |
> | Our face reconstruction (SAR) | 75.80 | 76.26 | 80.63 | 75.98 |
>
> As the results on this dataset show, the performance of the face recognition model and also our face reconstruction method is comparable for different age protocols in the AgeDB dataset. Therefore, it is reasonable to assume that facial attributes such as wrinkles may not be completely encoded in the facial templates, or even if there is some level of information, changing these attributes does not significantly change the facial templates.
>
> **Reply to Questions 4:**
> Following the reviewer's suggestion, we prepared a table in our `general response` and further described the adversary's knowledge and difficulty of each attack scenario. We will include this table in the final version of the paper.
> We would like to also mention that in research on the security and attacks against AI systems, evaluations are not often limited to whitebox/blackbox scenarios (against the same system), and the transferability of attack needs to be investigated to evaluate the robustness of generated samples by an adversary. However, to our knowledge, the transferability of the reconstructed face images in template inversion attacks has not been investigated in the literature. As a matter of fact, the transferability of reconstructed face images can lead to a critical threat where the adversary can use the reconstructed face image to enter another system.

---

> > ### Comment · Reviewer_S2tQ · 2023-08-22
> >
> > Thank the authors for their responses. I don't have any more questions. I hope the mentioned issues can be addressed in the final version. I will keep my final score.

---

> > > ### Author Response · Authors · 2023-08-22
> > > **Reply by Authors**
> > >
> > > We thank the reviewer for their feedback. We are happy that we could answer all reviewer's questions in our rebuttal. We will definitely include our new analyses reported in our rebuttal in the final version of the paper to address the mentioned issues. We appreciate the reviewer for their time in reviewing our paper and for the valuable comments, which helped us improve the quality of our paper.

---

### Author Rebuttal · Authors · 2023-08-09

We thank all reviewers for their time and valuable comments.
We tried to address point-by-point the concerns raised by the reviewers in individual responses.
For simplicity, we use the following numbers to refer to each reviewer in our responses:
- R1: Reviewer S2tQ
- R2: Reviewer T7AV
- R3: Reviewer RLF3
- R4: Reviewer pmB2
- R5: Reviewer dWCe

There are also some common comments between reviewers, which we reply to in this general response and refer to it in individual responses:

**Comparison of the Attack Scenarios:**
R1 and R5 suggested providing a comparison of the attack scenarios defined in our paper.
Following the reviewers' suggestion, we prepared Table 1 reported in our rebuttal pdf (attached) to further describe the adversary's knowledge and difficulty of each attack scenario. We will include this table in the final version and provide more descriptions to compare different attacks defined in our paper.
We would like to highlight that, to our knowledge, this is the first work that comprehensively evaluates the transferability of the reconstructed face images in template inversion attacks.
As a matter of fact, the transferability of reconstructed face images can lead to a critical threat where the adversary can use the reconstructed face image to enter another system in which the same user is enrolled.
However, to our knowledge, the transferability of the reconstructed face images in template inversion attacks has not been investigated in the literature.


**Using a Different Face Generator Network:**
R1, R2, R3, and R5 asked regarding the rationale behind the choice of StyleGAN and suggested exploring alternative synthesis networks.
StyleGAN is one of the most popular face generator models in the literature. It has available source code (and pretrained models), and several works have been using StyleGAN in different research problems.
However, our method can also be used with other face generator networks. As a new experiment, we use StyleSwin (CVPR, 2022), which is another face generator model based on transformers. Figure 1 in our rebuttal pdf (attached) shows the reconstructed face images from ArcFace templates using StyleSwin in our method instead of StyleGAN. We used a similar mapping network and learned a mapping from facial templates to the intermediate latent space of StyleSwin. As these results show, our method can also be used with other face generator networks. We will add these results to the final version of the paper too.

[StyleSwin] Zhang et al., "StyleSwin: Transformer-based gan for high-resolution image generation." CVPR 2022.

**Analysis of the Important Features in the Reconstructed Face Images:**
R1 raised the question of *what features between the template and synthetic images fool the recognition system* and suggested providing an  analysis with visualization to investigate why (and where) the attack works.
To answer this question raised by R1, we conducted a new experiment to explore what important information is encoded in the facial templates, and what features between the template and synthetic images fool the face recognition system. To this end, we applied the Grad-Cam algorithm using the face recognition model on the reconstructed face images to see which areas of the reconstructed face images are important and cause the facial templates of our reconstructed face images to be close to the original facial templates. Sample results of applying the Grad-Cam algorithm on our reconstructed face images are shown in Figure 2 of our rebuttal pdf file. As the results in this figure show, important areas that cause the reconstructed face images to have similar templates to the original ones correspond to areas such as eyes, nose, lips, etc. In particular, the area around the eyes seems to be the most important part in most of the reconstructed face images. These results also show that the general shape of the face (e.g., thin or chubby face), hairs, textures, etc., are not often necessarily important in the reconstructed face images, and thus, we can also conclude that these attributes are not well-encoded in the templates extracted by face recognition models. We will add this analysis to the final version too.

[Grad-Cam] Selvaraju, et al. "Grad-cam: Visual explanations from deep networks via gradient-based localization", ICCV 2017.


Finally, we would like to kindly invite all reviewers to further discussions should they still have any doubts or concerns.

---

> ### Author Response · Authors · 2023-08-15
> **Application of the proposed method for different inputs/outputs of face recognition models**
>
> We would like to append our explanations and experiment in our rebuttal to R5 to our general response to all reviewers. R5 raised the concern that we use three different face recognition models in our problem formulation but have an implicit assumption that the inputs and outputs of those models are the same. Therefore, R5 suggested more analyses for the cases when the inputs and outputs are different. However, our problem formulation and our proposed method, in fact, can be used in cases where the inputs/outputs of those models are different, as described hereafter. R4 also had concerns about the application of our method if the face recognition models require different face alignment methods, which we believe can be addressed by the same answer in our rebuttal to R5 as follows:
>
> **Application of our method for different inputs/outputs of face recognition models:**
> While we use three different face recognition models in our problem formulation, since these models are applied in separate stages, there is no issue if the inputs and outputs (e.g., pre-processing steps or dimensions) be different in each of these face recognition models.
> For differences in inputs (face images), because each of these models is applied independently on the given face image, the required pre-processing can be considered within the function of the face recognition model in our problem formulation. For differences in outputs (face templates), since the facial templates extracted by each model are compared to facial templates extracted by the same model, there is no conflict in the dimensions. The only point to be noted is that the input of our mapping network should have the same dimension as the templates of $F_{databse}$.
>
> Let us consider the complete pipeline of our problem formulation as depicted in Figure 2 of the paper. The first face recognition model $F_\{database}$ uses its own pre-process and extracts facial templates from face images captured by the camera of the face recognition system (from which the template is leaked). These facial templates (extracted from $F_\{database}$) are then used as input to our face reconstruction model. Therefore, the input of our mapping should have the same dimension as templates of $F_\{database}$. In any case, the output of the face reconstruction network is a high-resolution ($1024\times1024$) face image, regardless of the dimension of the input facial template. During training, the generated high-resolution face image is first pre-processed as required by $F_\{loss}$ (i.e., normalised, resized and aligned based on coordinates required by $F_\{loss}$), and the extracted templates are compared with templates of the original image extracted from $F_\{loss}$ (with the required pre-processing for $F_\{loss}$). During inference, however, the generated high-resolution face image is pre-processed as required by $F_\{target}$. Therefore, there is no conflict in the inputs/outputs in our pipeline.
>
> In our experiments reported in the paper, all face recognition models except Swin take input with  $112\times112$ resolution. However, the Swin model takes input with  $224\times224$ resolution.
> However, we acknowledge that the dimensions of facial templates extracted by all face recognition models in our experiments are similar and equal to 512.
> To show that our method can also be used in case of different dimensions of facial templates and to showcase another face recognition model with different pre-processing, as a new experiment, we used a new model (VGGFace) with **different dimensions of facial templates** (2048-dimension) and **different input image resolution** ($224\times224$) which has a **different normalization** as well as **different landmark coordinates for face alignment**. We used ArcFace as our $F_\text{loss}$ and evaluated the reconstructed face images in attacks against different face recognition systems (as $F_{target}$) on the LFW dataset:
>
> |  | ArcFace | ElasticFace| Swin |
> |---|---|---|---|
> | $\text{FMR}=10^{-2}\%$ | 92.92 | 93.10 | 83.97 |
> | $\text{FMR}=10^{-2}\%$ | 86.61 | 82.39 | 72.89 |
>
> As the results in this table show, our proposed method can be applied in the case where the inputs/outputs of face recognition models in our problem formulation ($F_{database}$, $F_{loss}$, $F_{target}$) are different, and still achieves high success attack rate against face recognition systems with different inputs/outputs.
>
> We will **add this explanation and experiment to the final version** to show the application of our method for such cases where the inputs/outputs of face recognition models are different.

---

### Decision · Program_Chairs · 2023-09-21

**Decision:**

Accept (poster)

**Comment:**

This submission proposes a method for realistic face image reconstruction from facial templates within a face recognition system. The advantage of this method is designed to work in both white-box and black-box cases. The submission received four borderline accepts and one borderline reject. The AC noticed that the negative comment from T7AV is very short and without details on weakness of this submission while the other four reviewers reach a consensus of acceptance. After reading the submission, comments, and rebuttals, the AC recognized the merits of this submission, and noticed that the minor weaknesses are well addressed. Therefore, the AC suggests accepting this submission.